

# Simulation of ash clouds after a Laacher See-type eruption

Ulrike Niemeier [1], Felix Riede [2], and Claudia Timmreck [1]

[1]The Atmosphere in the Earth System, Max Planck Institute for Meteorology, Bundesstr. 53, 20146 Hamburg, Germany
[2]Department of Archaeology and Heritage Studies, Aarhus University Moesgård, 8270 Højbjerg, Denmark

**Correspondence:** Ulrike Niemeier (ulrike.niemeier@mpimet.mpg.de)

**Abstract.** Dated to ca. 13,000 years ago, the Laacher See (East Eifel Volcanic Zone) eruption was one of the largest mid-latitude Northern Hemisphere volcanic events of the Late Pleistocene. This eruptive event not only impacted local environments and human communities but also NH climate. We have simulated the evolution of the fine ash and sulfur cloud of an LSE-type eruption under present-day meteorological conditions that mirror the empirically known ash transport distribution as

derived from geological, palaeo-ecological and archaeological evidence linked directly to the Late Pleistocene eruption of the Laacher See volcano. This evidence has informed our experimental set-up and we simulated corresponding eruptions of different injection altitudes (30, 60 and, 100 hPa) with varying emission strengths of sulfur and fine ash (1.5, 15, 100 Tg $SO_2$) and at different days in spring. The chosen eruption dates were determined by the stratospheric wind fields to reflect the empirically observed ash lobes. While it proved difficult to replicate the meteorological conditions that likely prevailed

13,000 years ago, our novel simulations suggest that the heating of the ash plays a crucial role for the transport of ash and sulfate. Depending on the altitude of the injection, the volcanic cloud begins to rotate one to three days after the eruption. The rotation, as well as the additional radiative heating of the fine ash, adds a southerly component to the transport vectors. This ash cloud-generated southerly migration process may at least partially explain why, as yet, no Laacher See tephra has been found in Greenlandic ice-cores. Sulfate transport, too, is impacted by the heating of the ash, resulting in a stronger transport to

low-latitudes, later arrival of the volcanic cloud in the Arctic regions and, a longer lifetime. Our models throw new light on the likely behaviour of the ash cloud that darkened European skies at the end of the Pleistocene, and serve as significant input for scenarios that consider the risks associated with re-awakened volcanism in the Eifel.

## 1 Introduction

The large VEI= 6/M= 6.2 explosive eruption of the Laacher See volcano dated to c. 13,000 yrs BP (Reinig et al., 2020) marks the end of explosive volcanism in the now dormant East Eifel Volcanic Zone (Germany). It was amongst the largest Late Pleistocene volcanic events in the Northern Hemisphere (NH) and has previously suggested to have temporarily impacted not only local environments (Baales et al., 2002), but also regional NH climate (Graf and Timmreck, 2001), as well as human communities even at some distance (Riede, 2008; Blong et al., 2018). It has also been suggested repeatedly - most recently





by Baldini et al. (2018) - that the eruption may in fact be implicated in the onset of the Greenland Stadial 1 cold spell that significantly interrupted the general warming trend of the Last-Glacial-Interglacial Transition and which led to the Younger Dryas ecological deterioration. The latter hypothesis is contested, however, due to uncertainties related to the dating of the Laacher See eruption (LSE) itself (see, Bronk Ramsey et al., 2015; Reinig et al., 2020) and the difficulty of linking this eruption conclusively to the Greenlandic ice-cores (e.g., Abbott and Davies, 2012), where a clear chemical signal or actual tephra shards from this eruption remain elusive.

Rather detailed reconstructions of the Laacher See eruption dynamics have been proposed (e.g., Schmincke et al., 1999; Bogaard and Schmincke, 1985; Schmincke, 2010). The eruption might have lasted several weeks possibly even months, most likely with an initial short (about 10 h) and intense phase, followed by a later explosive phase interspersed with eruption activity of varying intensity. Finds of plant macrofossils and animal tracks embedded in the proximal fallout of the eruption have revealed some important details: Finds of leaves and the tracks left by young animals indicate a late spring/early summer date of the eruption (Baales et al., 2002); this seasonal determination is supported by highly resolved palaeoecological observations in, for instance, varved lake sediments that indicate fallout ash deposition after the formation of the winter layer but also prior to the deposition of sediments associated with summer (e.g., Merkt and Müller, 1999; Hajdas et al., 1995). The same animal prints also suggest that the eruption lasted long enough and was characterised by at least some subdued phases for rain to fall and for animals to make their way through the ash-covered landscape. The distal ash distribution is also interesting in this regard. The LSE shows an unusual, two-lobed pattern with deposits belonging to a massive primary lobe stretching over north-east Germany and the Baltic Sea towards north-west Russia, and a secondary lobe leaving deposits to the south of the volcano towards the Alps (Riede et al., 2011; Reinig et al., 2020). This two-lobed fallout distribution also suggests that (i) the eruption phases were of very different intensity with ejecta reaching different heights dominated by different wind directions, and/or that (ii) the duration of the eruption was long enough for the dominant wind directions to shift significantly. The eruptive phases are, following Schmincke (2010), divided into Lower Laacher See Tephra (LLST, first Plinian stage) and Middle Laacher See Tephra (MLST A, B, C; second Plinian stage), and a late and generally less explosive Upper Laacher See Tephra (ULST). These data are used to constrain the novel simulations presented here.

Considerable advances in the modelling of volcanically-induced climatic forcing of NH mid-latitude eruptions have recently been made (Toohey et al., 2019), and these warrant renewed attention to the Laacher See eruption's potential influence on NH climate. The rich volcanological detail associated with this Late Pleistocene eruption facilitate a better understanding of its interaction with potential meteorological conditions shaping its ash and aerosol dispersal. These interactions, in turn, are important for not only addressing the enigmatic absence of this eruption in the ice-core records as well as for unraveling to what climatic effects associated with the eruption may have influenced the human responses observable in the archaeological record. These responses range from regional depopulation to migration and cultural florescence but it remains contested as to whether the reduction in ecosystem services due to tephra fall or the climatic impacts of the eruption shaped these responses (Riede, 2017; Blong et al., 2018). In addition, recent research is also revisiting the LSE as a model worst-case scenario (cf., Aspinall and Woo, 2019) for considering the damages, costs and surge capacity requirements of contemporary society to a potential Laacher See-type eruption (Leder et al., 2017; Riede, 2017). The practice of using historical eruption data to constrain future





emergency planning is well-established in municipalities plagued by active volcanism (e.g. Vesuvius: Mastrolorenzo et al.,
     2006; Zuccaro et al., 2008; Martin, 2020). A number of national governments also use, for instance, Laki-type eruptions to
     derive so-called Realistic Disaster Scenarios (Mazzorana et al., 2009) for the long-range aerosol-mediated impacts of NH
     volcanism on contemporary societies (Schmidt et al., 2011; Sonnek et al., 2017). Developing robust models for a Laacher See-
     type eruption and its potential impact would thus not only facilitate a further exploration of the impact of the actual eruption
on past communities but also its use as a Realistic Disaster Scenario that addresses the combination of ash- and aerosol-driven
     impacts as well as critical issues of communication, cross-border coordination, migration and infrastructural damage beyond
     the proximal impact zone (Donovan and Oppenheimer, 2018).

     Against this background and building on much earlier work by Graf and Timmreck (2001), we here present new simulations
     of a Laacher See-type eruption under present-day climatic conditions. We do not attempt to reconstruct the climatic impact of
the LSE itself as it occurred during the Late Pleistocene, but draw on the available volcanological and palaeoecological proxy
     data to realistically constrain our simulation (Section 2). We thus use the LSE as a shorthand for a substantial, highly explosive
     Plinian NH mid-latitude (Laacher See-type) eruption. We present significant new insights into ash transport and deposition
     (Section3.1) as well as the role of fine ash for the transport of sulfate and for the magnitude of climatic forcing associated
     with such a mid-latitude eruption (Section 3.2). While our simulations do not attempt to reconstruct the likely climatic impacts
of the Late Pleistocene LSE itself, our study does have implications for our understanding of that eruptive event and its past
     socio-ecological consequences.

## 2   Model and Simulations

### 2.1   Model description

     The simulations for this study were performed with the middle atmosphere (GCM) MAECHAM5 (Giorgetta et al., 2006).
MAECHAM5 was applied with the spectral truncation at wave-number 63 (T63), a grid size of about $1.8° \times 1.8°$, and 95
     vertical layers up to 0.01 hPa. The model solves prognostic equations for temperature, surface pressure, vorticity, divergence,
     and phases of water. To simulate the evolution of a volcanic cloud, MAECHAM5 was interactively coupled to the prognostic
     modal aerosol microphysical model HAM (Stier et al., 2005), which calculates the sulfate aerosol formation including nu-
     cleation, accumulation, condensation and coagulation, as well as its removal processes by sedimentation and deposition. The
initial conversion of $SO_2$ into $H_2SO_4$, is simulated with a simple stratospheric sulfur chemistry scheme, which is applied above
     the tropopause (Timmreck, 2001; Hommel et al., 2011).

     Ash particles are relatively large and sediment quickly out of the stratosphere, usually already during the first days after the
     eruption. We simulate fine ash with one mode only and do not take into account large ash particles that fall out swiftly and
     usually very close to the eruptive centre. For the fine ash mode we assume a standard deviation of $\sigma = 1.8$, a density of 2400
$kg\,m^{-3}$, a wet mean radius of $r_{wet} = 2.43\,10^{-6}$ m and an effective radius of $r_{eff} = 4.16\,10^{-6}$ m similar to the simulation of
     the June 1991 Pinatubo eruption by Niemeier et al. (2009). The radiative direct effect of fine ash and sulfate aerosol is included
     for both solar (short wave, SW) and terrestrial (long wave, LW) radiation, and coupled to the radiation scheme of ECHAM.





We calculate the aerosol radiative forcing at top of the atmosphere (TOA) with a double call of the radiation, once with and once without aerosols. The fine ash and sulfate aerosols dynamically influence the resulting processes via temperature changes caused by absorption of near-infrared and LW radiation. This model has already been successful applied for the simulation of recent and past large volcanic eruptions (e.g., Niemeier et al., 2009, 2019; Toohey et al., 2016, 2019), and further model details are described in Niemeier et al. (2009) and Niemeier and Timmreck (2015).

For our simulations only natural sulfur emissions are taken into account. Land-sea mask, sea surface temperature (SST) and sea ice are prescribed for present-day conditions. SST and sea-ice are set to climatological values (Hurrell et al., 2008), averaged over the period 1950 to 2000. Although our boundary conditions are not representative for the SST during the Late Pleistocene, we assume that their impact on our results are small especially as the eruption itself almost certainly caused a strong disturbance in stratospheric flow pattern. It is likely, however, that stratospheric circulation was somewhat different in spring as stratospheric dynamics respond to the conditions of Arctic sea ice cover (Jaiser et al., 2013), which certainly was different 13,000 years ago. This might play a role for determining the specific day of the eruption as discussed in Section 2.2.2.

## 2.2 Simulations

### 2.2.1 Source parameters

An eruption history of the LSE has been reconstructed and described in detail by Schmincke et al. (1999) whom we follow here for setting the basic eruption parameter ranges. We focus here on a Laacher See-type eruption which could reproduce the observed two-lobed pattern. Hence, we consider in our simulations two eruptions phases, a first ten hour-long explosive eruption phase corresponding to the LLST which is connected to the north-eastern lobe and a second three hour-long phase which represents the less substantial southern lobe corresponding to eruption phase MLST-C. The eruption is initialized over the grid box where the Laacher See is located (50.24°N, 7.16°E).

An estimation of the amount of fine ash corresponding to the two historic ash lobes released into the stratosphere is difficult. Only limited particle size data for the distal Laacher See tephra are available (Riede and Bazely, 2009). These data, however, are heterogeneously generated and not directly comparable to present-day instrumental observations, nor are they representative of the LSE on the whole – and hence not appropriate as modelling input. While comparable in Volcanic Explosivity Index (Newhall and Self, 1982), the calculated magnitude of the LSE ($M = 6.2$) is slightly greater than the 1991 Mt. Pinatubo eruption ($M = 6$), with 1 to 10 $\mathrm{km}^3$ erupted tephra mass (Textor et al., 2003) or 20 $\mathrm{km}^3$ of ejecta (Baales et al., 2002). Yet, the amount of fine ash that reached the stratosphere is likely much smaller. Pinatubo simulations (Niemeier et al., 2009) indicate that a reasonable number for fine volcanic ash particles reaching the stratosphere is 1% of the total emitted ash to align with satellite observations (Guo et al., 2004). Previous studies (Textor et al., 2003) have suggested an eruption rate of roughly $4 \cdot 10^8$ $\mathrm{kg\,s}^{-1}$ for the explosive LLST and MLST-C phases. Given this eruption rate for total ash, an approximate duration of the eruption phase of 10 hours and the 99% reduction for fine stratospheric ash we estimate a total erupted mass of fine ash in the stratosphere of 150 Tg for the first eruption phase.



The amount of sulfur released of the ancient LSE is not very well known, and estimates span a range of almost three orders of magnitude (Baldini et al., 2018). We therefore performed simulations with three different $SO_2$ emissions: 1.5, 15 and, 100 Tg ($SO_2$), for the LLST eruption phase to include the range of most likely estimates (Textor et al., 2003) in our study. We define here the 15 Tg ($SO_2$), which was also used in the study of Graf and Timmreck (2001) together with 150 Tg of fine ash as our reference emission scenario. The ratio of erupted mass of $SO_2$ to fine ash (1:10) is assumed to be constant in all the
simulations discussed here.

No information are available for the injection profile. Observations of more recent eruptions and numerical simulations suggest a separation of ash and sulfate in the eruptive cloud (Schneider et al., 1999; Holasek et al., 1996; Prata et al., 2017) with a lower neutral buoyancy height for fine ash than for sulfur. In the absence of of pertinent data we assume for our simulations an injection profile for $SO_2$ and fine ash which has been derived from satellite observations and been used for the
simulation of the 1991 Mt Pinatubo eruption (Niemeier et al., 2009). The 1991 Mt. Pinatubo eruption was a tropical one, mid to high latitudes eruptions might reach not so high into the stratosphere. We therefore consider also two scenarios with lower injection altitudes for $SO_2$ 60 and, 100 hPa and 80 and, 120 hPa for fine ash, keeping the vertical offset between the sulfur and ash emission layers constant.

For the second eruption phase we conservatively assume one third of the $SO_2$ and fine ash of the first eruption phase has
been injected based on the respective tephra volumes of the proximal LLST and MLST-C deposits. We also adopt the injection profile of the 2nd eruption phase assuming with 220 hPa for $SO_2$ and 240 hPa for fine ash lower injection altitudes compared the 1st phase which just reached the lowermost stratosphere. An overview of the different LSE simulations is given in Table 1.

### 2.2.2    Eruption day

The distribution and subsequent evolution of the volcanic cloud depends on the meteorological conditions of the stratosphere
at the time of the eruption (Jones et al., 2016; Toohey et al., 2019). Ash deposition patterns reflect the long-range transport of volcanic ash and hence the meteorological situation in the lower stratosphere at the time of the eruption. The palaeontological (botanical and trace-zoological) evidence preserved in the proximal LSE ash deposits offers strong indications of a late spring/early summer date of the eruption, although it is also worth noting that the environs of the Late Pleistocene were of a non-analogue nature, i.e. the presence of particular plants and their phenology or of particular animals should not be uncriti-
cally mapped onto present-day conditions. It is therefore almost impossible to simulate an ash deposition in a numerical model that matches exactly an empirically known one, not least a deposition pattern as complicated as that of the actual LSE. For a present-day eruption, observational data could be used together with nudging (e.g. ECMWF analysis data), to push the model into a state that is similar to the weather and wind situation at the eruption day. This is not possible for ancient eruptions. Therefore, we used the known tephra lobe deposition as a prior lead for the conditions in the stratosphere during the LSE in
the Late Pleistocene: south-westerly wind causing transport to the Baltic Sea for the first explosive eruption phase (LLST) and northerly wind for the second explosive eruption phase (MLST-C). Winds in the stratosphere vary strongly by season. During summer at an altitude of 30 hPa, easterly winds between 50°N and 60°N are dominant and westerly winds during winter.





**Table 1.** Overview of the different LSE simulations. The emitted mass of the second eruption is 1/3 of the first eruption for $SO_2$ and ash, respectively. The first number of the injection altitude is the altitude of the $SO_2$ injection, the second of the ash injection. The duration of the first phase (LLST) was assumed to be 10 hours and the second phase (MLST-C) 3 hours.

| | First phase (LLST) | | | | Second phase (MLST-C) | | | |
|---|---|---|---|---|---|---|---|---|
| No | Emission | Fine ash mass/ particle number | Injection altitude | Date | Emission | Fine ash mass/ particle number | Injection altitude | Date |
| | [Tg $SO_2$] | [Tg ash]/[part.] | [hPa] | | [Tg $SO_2$] | [Tg ash]/[part.] | [hPa] | |
| LSE1 | 15 | 150 / 2.2 $10^{23}$ | 30/50 | May 7th | 5 | 50 / 7.32 $10^{22}$ | 220/240 | June 20st |
| LSE2 | 15 | 150 / 2.2 $10^{23}$ | 60/80 | May 7th | 5 | 50 / 7.32 $10^{22}$ | 220/240 | June 20st |
| LSE3 | 15 | 150 / 2.2 $10^{23}$ | 100/120 | May7th | 5 | 50 / 7.32 $10^{22}$ | 220/240 | June 20st |
| LSE4 | 1.5 | 15 / 0.22$10^{23}$ | 30/50 | May 7th | 0.5 | 5 / 0.732$10^{22}$ | 220/240 | June 20st |
| LSE5 | 100 | 1000 / 14.8$10^{23}$ | 30/50 | May 7th | 33.3 | 333.3/ 49$10^{22}$ | 220/240 | June 20st |
| LSE6 | 15 | 150 /2.2 $10^{23}$ | 30/50 | May 15th | 5 | 50 /7.32$10^{22}$ | 220/240 | June 20st |
| LSE7 | 15 | 150 /2.2 $10^{23}$ | 30/50 | May 22nd | 5 | 50 /7.32$10^{22}$ | 220/240 | June 20st |
| LSE8 | 15 | 0 / 0 | 30/- | May 7th | 5 | 0 / 0 | 220/- | June 20st |
| LSE9 | 15 | 0 / 0 | 100/- | May 7th | 5 | 0 / 0 | 220/- | June 20st |
| LSE10 | 15 | 0 / 0 | 30/- | May 15th | 5 | 0 / 0 | 220/- | June 20st |
| LSE11 | 15 | 0 / 0 | 30/- | May 22nd | 5 | 0 / 0 | 220/- | June 20st |

During spring, and after the break-down of the polar vortex, the situation can be more complex with local low or high pressure systems. It is these that would allow a transport of the ash from the East Eifel towards the Baltic Sea.

We performed a control simulation without volcanic eruption and checked the meteorological situation in the stratosphere in spring of three different years. In May of one year we found a situation similar to the assumed conditions at the LSE. Figure 1 shows the flow pattern at 48 hPa, close to our reference injection height for different days in May of this specific year. Our model shows strong easterly winds from late May onwards, for instance, on May 22nd (Figure 1, d). We therefore selected early May for model initialisation, even if this is not in full agreement with the palaeontological evidence. The best agreement of the

spatio-temporal distribution between simulated and observed ash deposits was found for May 7th as starting day for the LLST eruption phase (LSE1, see definition of simulations in Table 1). Figure 2 b) shows the known distribution of the historical LSE ash lobes (all sites in black, sites with assumed LSE ash remains of LLST and MLST-C in brown and grey). We performed two additional simulations with explosive eruption events on May 15th (LSE6) and May 22nd (LSE7) to highlight the impact of the dynamic state of the stratosphere on the dispersion of the volcanic cloud. Additionally, we performed simulations without

the injection of fine ash for the three injection dates. This allows us to use this small ensemble to discuss the rile of fine ash on tracer distribution and transport.

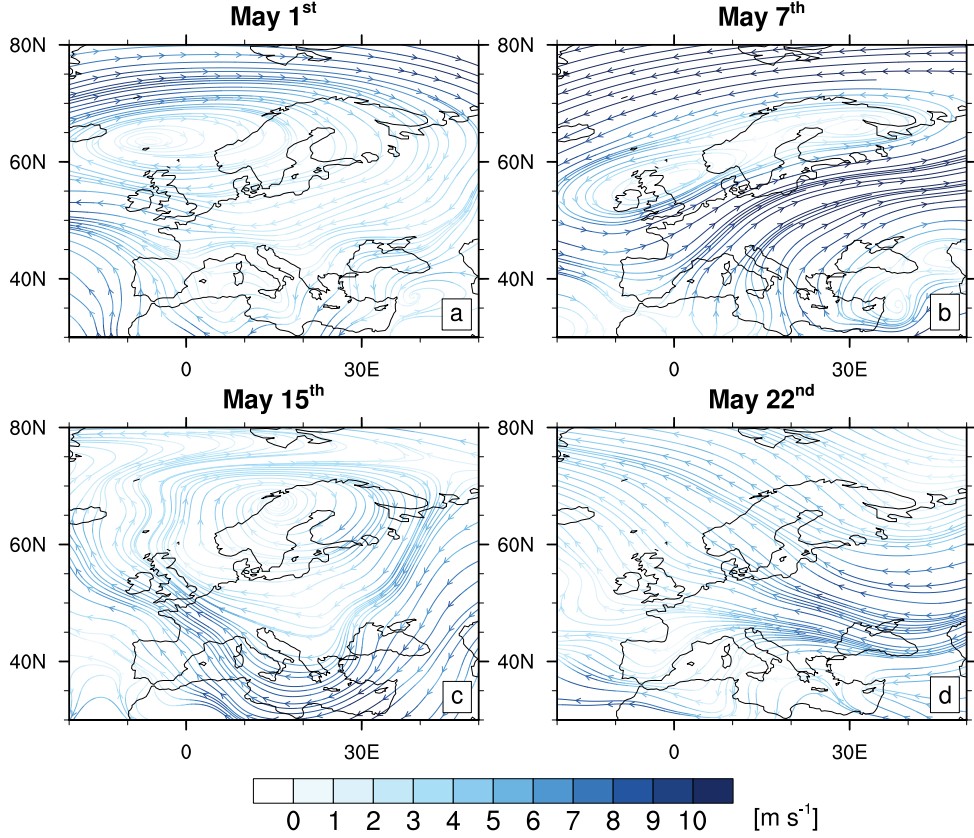

**Figure 1.** Streamlines of the undisturbed zonal wind $[\mathrm{ms}^{-1}]$ of the control simulation over Europe at 48 hPa at different days in May.

In order to determine a day for the MLST-C eruption phase, we continued the simulation after the first explosive eruption bout but without a second explosive eruption phase. This simulation provided the basis for identifying a date for the MLST-C phase with transport to the south/south-west (see grey markers in Figure 2 b). The results that best match the simulated ash
deposits to those known empirically were obtained for June 20th. This day was used for all simulations despite the fact that after the LLST, the dynamic conditions changed (Figure A1) and the deposition structure of the second explosive eruption phase could only be reproduced in LSE1.

## 3   Results

### 3.1   Simulation of fine ash

Volcanic ash plays an important role in the very early phase after an eruption. Ash particles are relatively large and sediment quickly out of the stratosphere usually already during the first days after an eruption, although some very fine ash particles can



remain in the stratosphere for longer (Vernier et al., 2016). Once in place, ash clouds are heated by absorption of solar radiation causing an additional vertical updraft. This heating occurs right after the eruption, before the substantial formation of sulfate aerosols.

### 3.1.1 Sensitivity to emission height and strength

Emission rate and altitude have a major impact on the deposition pattern of fine ash. In our study, the explosive eruption days, May 7th for the first phase and June 20th for the second, were chosen to simulate as closely as possible the empirically known tephra distribution of the LSE phases (Schmincke, 2010; Riede et al., 2011; Reinig et al., 2020). Figure 2 b) shows the currently know ash deposits of the LSE lobes. The transport towards the Baltic Sea after the first eruption is captured well in all simulations (Figure 2). LSE1 shows the main deposition closest to the Baltic Sea of all simulations. For LSE2 and LSE3, where $SO_2$ and ash are injected at lower altitudes, the maximum deposition occurs farther to the east. The distribution of deposited ash is more narrow but longer with a more pronounced eastward spread. The distribution of deposited ash in LSE4 is similar, but, due to lower injected mass, the absolute value is much smaller. The opposite is the case for LSE5. The main area of deposition is similar to LSE2 and LSE3, but the spread is much stronger and the ash deposits correspondingly cover a much greater area.

The estimated pattern of tephra distribution of the MLST-C phase, main deposition towards the south, is also well captured in LSE1. Ash deposition in model runs LSE2 and LSE3 shows a similar pattern, albeit with deposition occurring preferentially over the Adriatic Sea, and also over England and the North Sea. The southward distribution of fine ash deposition in LSE4 and LSE5 is very different. The absorption of radiation, mainly solar radiation, heats the layer of ash. The ash-induced heating changes the wind pattern in the stratosphere. These changes depend on the injection altitude and, more importantly, on the emitted mass. Consequently, the wind in the stratosphere is in different states on June 20th in all simulations (Figure A1) with the result that transport directions of the ash associated with the later MLST-C explosive phase differ substantially between model runs and in relation to the empirical benchmark of the Late Pleistocene eruption. The differences in transport after the first phase are more related to a direct impact of the heated ash cloud on the wind pattern in the stratosphere, as described in the next section.

Our results indicate almost no transport of ash to high latitudes, except in LSE5 characterised by a very strong eruption rate. Figure 2 indicates that transport of ash to higher latitudes depends on the injection height. LSE1 shows a small amount of deposited ash over Iceland, LSE2 even slightly further north. LSE3 shows no ash deposition north of 70°N. However, do note that our simulations represent only a single state of the atmosphere out of many possible ones. The chosen day reflects the observed ash deposits in central Europe. Thus, the winds reflect the historical wind conditions in central Europe. Further away from the eruption site the transport path of the tracer has been, most probably, different during the historical LSE.

### 3.1.2 Role of rotating ash cloud

The deposition pattern of ash of the LLST explosive eruption phase in May, with deposition along the Baltic Sea, shows a turn towards south in all cases. This feature is related to the heating of the ash due to absorption of solar radiation and the consequent



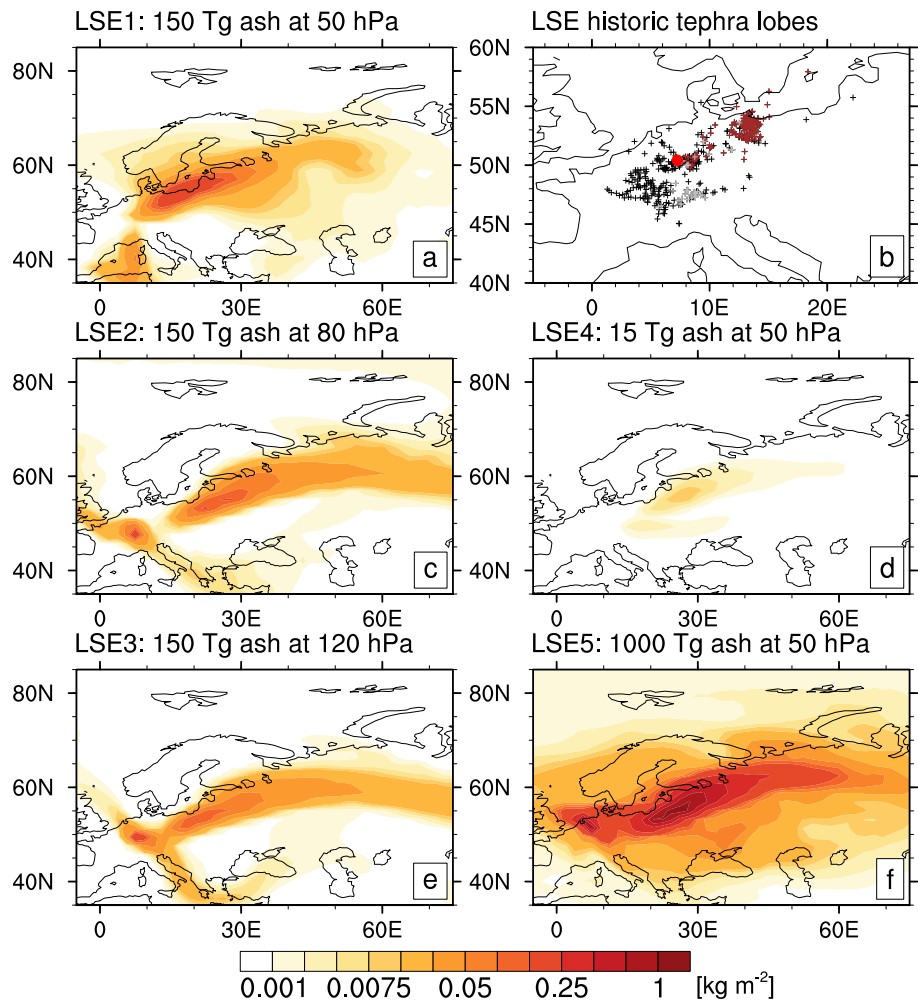

**Figure 2.** Deposition of fallout ash accumulated over May and June following the explosive eruption phases in simulations LSE1 to LSE5. Simulations with injection of 150 Tg fine ash at different altitudes (a,c,d), Simulations with two different injection rates, both at 50 hPa for the fine ash (d, f), and, b) currently know distribution of all Late Pleistocene tephra deposits (black) of the LSE (LLST, MLST, and ULST); LLST= brown, MLST= grey (Riede et al., 2011; Reinig et al., 2020). Note that many LST finds are not directly associated to any specific eruption phase. The red dot marks the eruptive centre.

impact on the stratospheric winds. The heated air causes a vertical updraft, a change of density and positive divergence due to expanding air at the top of the cloud (Baines and Sparks, 2005; Costa et al., 2013) where the vertical motion within the volcanic cloud turns into a horizontal outflow. Under the influence of the Coriolis force a right turn of horizontally expanding air and even an anti-cyclonic rotation of the heated volcanic cloud may develop. At night, the upper part of the cloud becomes colder. Without the heating of solar radiation, upward motion of the cloud ceases and sedimentation increases. Then the cloud





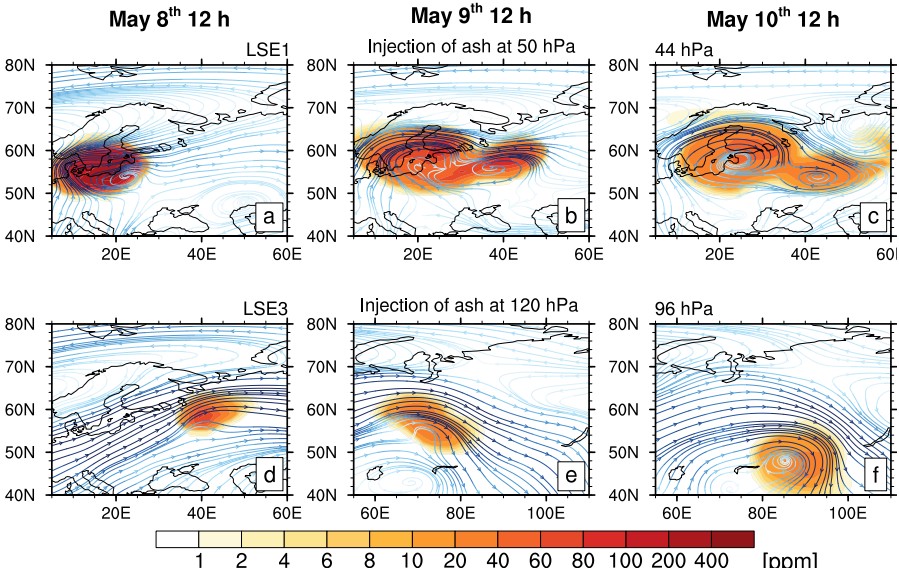

**Figure 3.** Ash concentration (shaded) and streamlines of the zonal wind for scenario LSE1 (top, a-c) and LSE3 (bottom d-f) at the 1st (left column) and 2nd (middle) and third (right) day after the 1st phase LLST. The ash is plotted in both scenarios in the second vertical level above the injection altitude, 44 hPa and 96 hPa, respectively. The color scale of the streamlines is similar to Figure 1 but represents 10 values between 0 and 20 ms$^{-1}$. Note the different area in Figures e) and f).

is no longer expanding, divergence becomes negative, and the anticyclonic motion is less pronounced but not breaking down. Without the radiative heating of the cloud, no rotation develops.

     Figure 3 shows the streamlines of the wind slightly above the eruption altitude, the area of positive divergence described above, for the three days after the first eruption phase of simulation LSE1 and LSE3. At the higher injection altitude (50 hPa) of LSE1, the ash cloud starts rotating shortly after the eruption while for LSE3 only a slight right turn of the flow is simulated.

The rotating ash cloud of LSE1 stays closer to the eruption site and is less strongly transported with the wind. The fast easterly transport of of ash in LSE3 is diminished at the third day after the eruption, when the ash cloud of LSE3 has risen and starts to rotate as well (May 10th). The vertical extension of the cloud is driven by the injection altitude. In LSE1, the vertical distance to the tropopause is larger, allowing a larger vertical extension of the cloud and stronger heating, and a longer lifetime. In LSE3 the ash cloud has to rise to higher altitudes before the rotation develops. In addition, the difference between the density of the

cloud and the density of the environment is larger at higher injection altitudes which may increase the velocity of the horizontal outflow. Previous work on the formation of an umbrella cloud (e.g. Baines and Sparks (2005); Costa et al. (2013)) does not discuss this aspect of ash cloud dynamics.

     This rotation of the volcanic cloud may explain the local maxima and uneven deposition of Laacher See tephra in the eruption's medial field in particular (see Riede et al. (2011)). Both simulation scenarios demonstrate how the heated ash cloud




impacts the flow and the dispersion of the cloud itself. The clockwise turn of the air masses hinders transport to the north and this mechanism could therefore also offer an explanation for the absence of LSE deposits in the Greenlandic ice-cores.

## 3.2   Impact of sulfate aerosols

### 3.2.1   Global distribution of sulfate burden

Sulfate aerosols have a longer lifetime than fine ash and elicit a stronger climate impact. The LSE is an extra-tropical eruption

and could, locally, have led to a stronger impact than a tropical eruption of the same size (Toohey et al., 2019). Following an extra-tropical eruption sulfate is mainly transported within the Brewer-Dobson circulation (BDC) to higher northern latitudes (Figure 4). The aerosols reach the high latitudes about one to two months after the eruption. Smaller amounts of sulfate reach the equatorial latitudes roughly two months after the eruption and with the transition to northern winter conditions cross the equator and are transported towards the southern high latitudes.

Meridional transport in the stratosphere depends on the injection altitude and is stronger within the lower stratosphere due to wave-induced turbulent structures. This has implications for the simulated sulfate transport in the LSE1 to LSE3 simulations, which differ in their emission profiles (Figure 4, a, c and e). In LSE1 — the simulation with the highest injection altitude — the volcanic cloud arrives later at the pole with less sulfate than in LSE2. Sulfur injection in 100 hPa, LSE3, causes the smallest burden but the aerosols stay longer in the stratosphere (Figure 4, e) which is related to smaller particles (Figure A2). Meridional

transport is stronger in the lower stratosphere which results in a faster dilution of the injected sulfur and consequently in smaller particles. This is in line with previous studies ((Toohey et al., 2019; Marshall et al., 2019)) that also show that effective radii of volcanic sulfate particles are smaller for an initial injection at 100 hPa compared to an injection at 30 hPa. Additionally, LSE3 shows the strongest transport to lower latitudes which reduces the local sulfur load and also the particle radii.

Decreasing the injection rate by a factor of ten (LSE4) and increasing it by a factor of seven (LSE5) reveals, again, the

non-linearity of our results. The calculated maximum burden values of eight (LSE4), 50 (LSE1) and, 370 $\mathrm{mg\,m^{-2}}$ (LSE5) do not reflect the factor of emission change (Figure 4). Likewise, one would expect a less pronounced burden increase between LSE1 and LSE5 compared to LSE4 and LSE1 as particle size should increase with increasing emission and, thus, the lifetime decrease. This shows the non-linearity of the sulfate evolution. The meteorological wind conditions are the same at eruption time in LSE1, LSE4 and, LSE5, but additional to non-linear microphysical processes, the impact on the stratospheric wind

pattern by the volcanic cloud is different for different emissions. In LSE4 the heating of ash is too small for a well developed rotating cloud and sulfate reaches high latitudes early. In LSE5 we see both fast transport to high latitudes as well as stronger transport into the tropics and southern hemisphere than in LSE1. However, the lifetime of the aerosols is notably similar in both simulations.

The monthly mean sulfate burden in May and June (Figure 5) reveals more details regarding the differences in transport

within the first two months after the eruption for those simulations with different injection altitudes. In May, the main transport occurs with easterly winds at 30 hPa (LSE1) and with westerly winds at 100 hPa (LSE3) and 60 hPa (LSE2, Figure A4). We argue that the rotating ash cloud impacts not only the transport of ash, but also of sulfur. Therefore, we discus also the impact

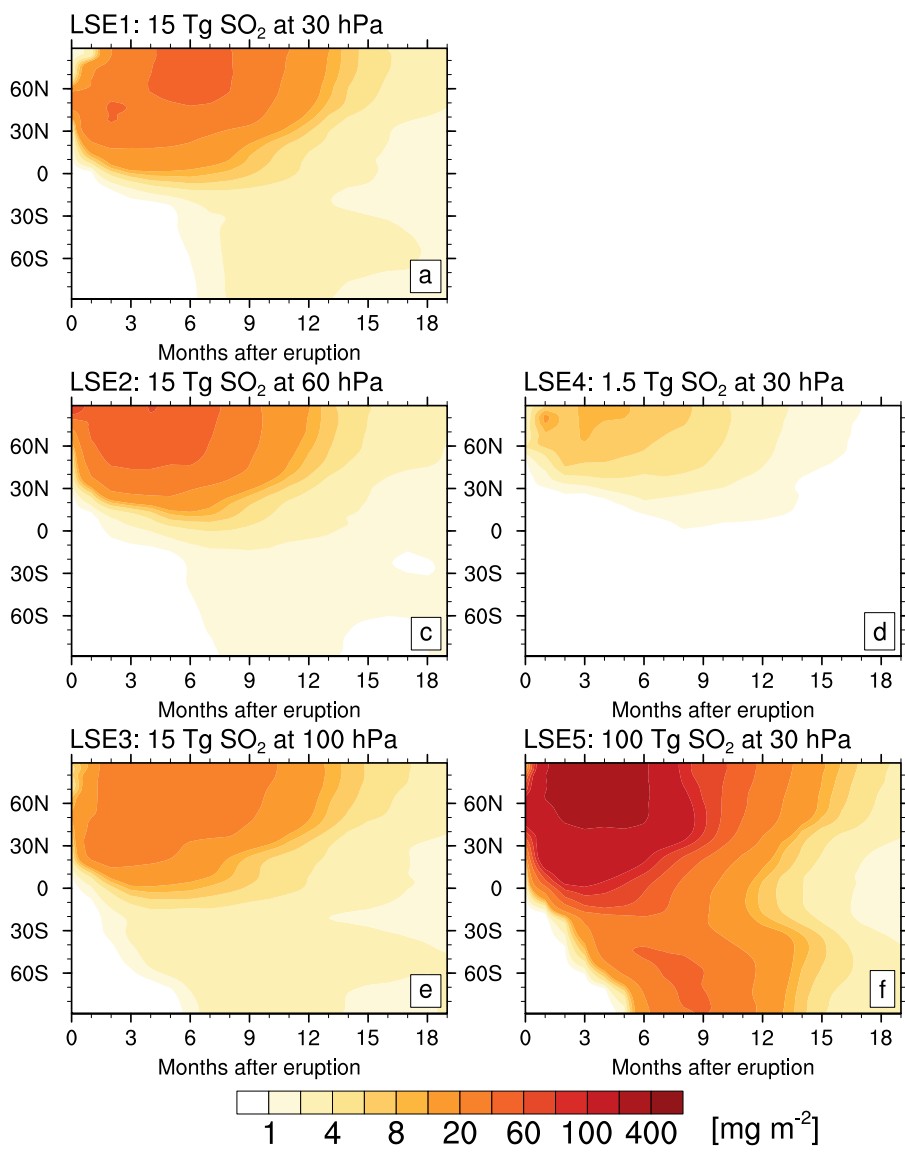

**Figure 4.** Hovmøller diagram of zonally averaged sulfate burden, vertical integral of sulfate, over a period of 1.5 years after the first eruption phase. Zero is in May, the month of the first eruption phase. Plotted values are 1, 2, 4, 8, 10, 20 $\mathrm{mg\ m^{-2}}$ etc.

of volcanic ash on the sulfate distribution. The results of May clearly illustrate the impact of the rotating volcanic cloud (Figure 5). The cloud starts to rotate on day one (LSE1), two (LSE2) and three (LSE3) after the eruption. This rotation slows down
the zonal transport as discussed for the deposition of fine ash in Section 3.1.2. Figure 5 shows a similar result for the sulfate transport. Results of LSE8 and LSE9, simulations without ash injection, show stronger horizontal transport in May (Figure 5)





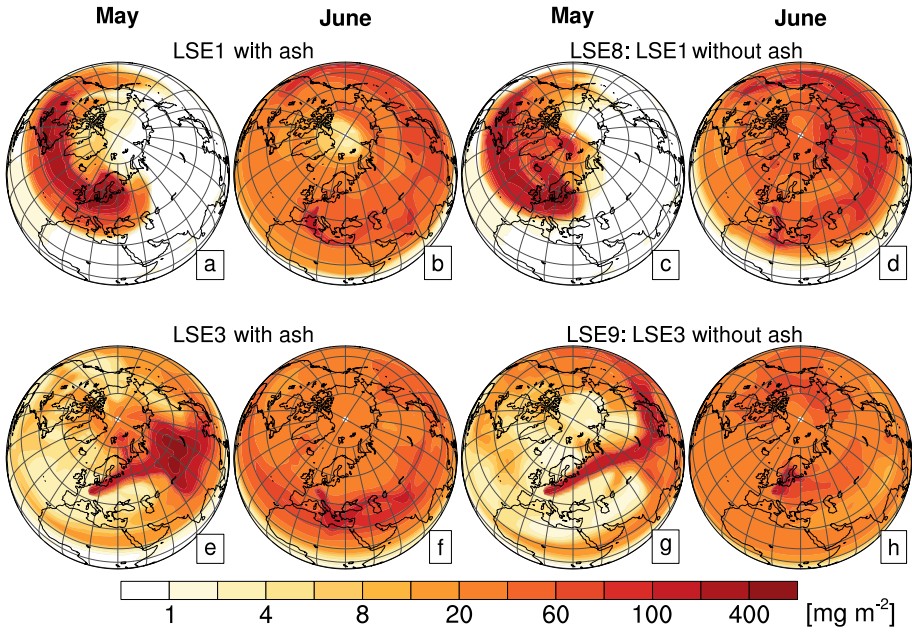

**Figure 5.** Monthly mean sulfur burden (SO₂ plus sulfate) with (a, b, e,f) and without (c, d, g, h) the injection of ash shortly after the eruption (May and June) for the scenarios LSE1, LSE3, LSE8 and, LSE9 with the same eruption rate but different injection altitude (30 hPa, a to d) and 100 hPa, e to h). Plotted values are 1, 2, 4, 8, 10, 20 mg m$^{-2}$ etc.

compared to LSE1 and LSE3. In LSE1, the rotation of the volcanic cloud keeps the SO₂ and sulfate aerosols over Scandinavia right after the eruption phase in May, while in LSE8 sulfate is transported to Greenland and Swalbard. In June, the sulfate is widespread over the northern hemisphere, with enhanced values following the cloud of the second explosive phase. In LSE1

only a very small amount of sulfate reached the pole, in contrast to the simulation without ash (LSE8). Comparing LSE3 to the simulation without ash (LSE9) the difference in transport is clearly related to the rotation of the volcanic cloud. With ash (LSE3) the cloud widens over Siberia in May were the cloud starts to rotate, which leads to a change in the transport pattern. Opposite in LS9, the cloud is transported straight over Siberia. The differences in ash deposition are small between LSE2 and LSE3, but the sulfate distributions are rather different. Transport is similar for LSE2 and LSE3 along the Baltic sea right after

LLST (Figures 5 e and A4 a). This changes with the onset of the rotation over Finland in LSE2 and over Siberia in LSE3. The consequence is a strong poleward transport in LSE2 but not in LSE3. Thus, the rotation slows down the zonal transport but widens the cloud. These examples show that the transport depends on details of injection rate and altitude which impact the flow pattern differently.

    In the 12 months following LSE8 (Figure 6, b), we observe a stronger transport to high latitudes and less equatorward

transport into the tropics compared to LSE1. This result is confirmed for simulation LSE3 and LSE9. Overall, the rotating ash





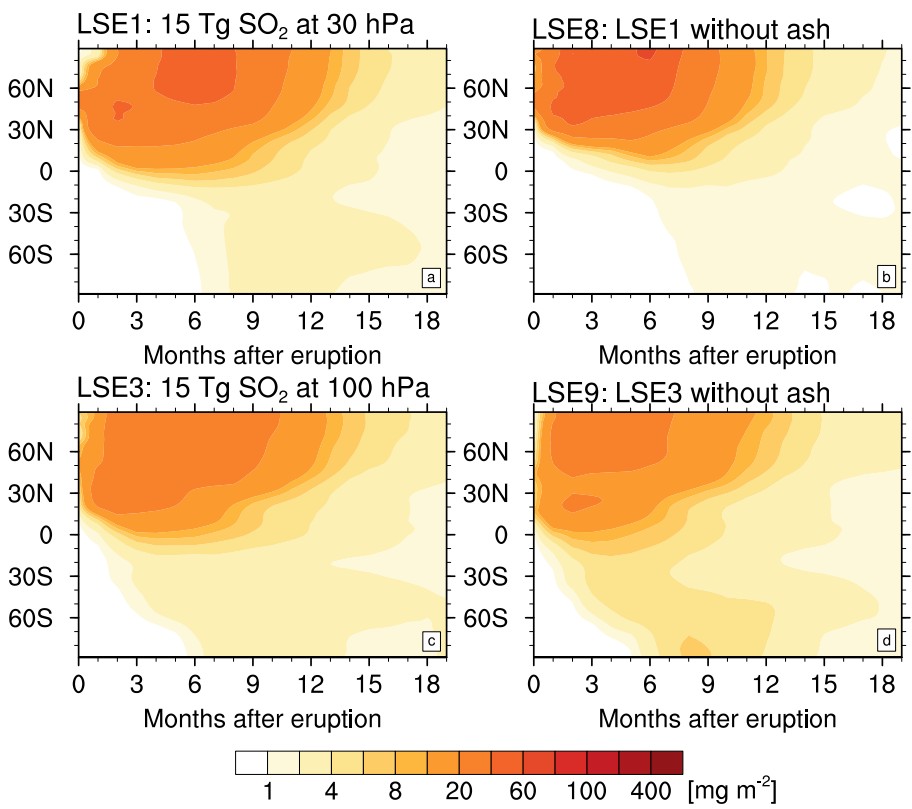

**Figure 6.** Hovmøller diagram of zonally averaged sulfate burden for an assumed LSE-like eruption over a period of 1.5 years after the first eruption phase. Left: Simulations with injection of fine ash a) LSE1 c) LSE3. Right: the corresponding simulations without the injection of fine ash, b) LSE8 and d) LSE9.

cloud adds a southern component to the transport. Additionally, the zonal mean heating rates indicate in LSE1 a stronger heating right after the eruption at 50°N, due to the presence of ash, as well as stronger heating and vertical lofting in the volcanic cloud at 30°N (Figure A3). Within ECHAM-HAM, the absorption in the near infra-red is important. Thus, the stronger equatorward transport with ash emissions results in a stronger heating as solar irradiation is stronger in mid- and low latitudes than at high latitudes.

### 3.2.2 LSE eruption later in May

We discussed earlier the conditions in the stratosphere during a specific eruption date, May 7th. Fixing the eruption date in this way allows us to match the ash lobes of the 13ka BP eruption. The consequence of this forced date-fixing is, however, that ensemble modeling was not possible. To mitigate this missing ensemble, usually necessary to take into account different states



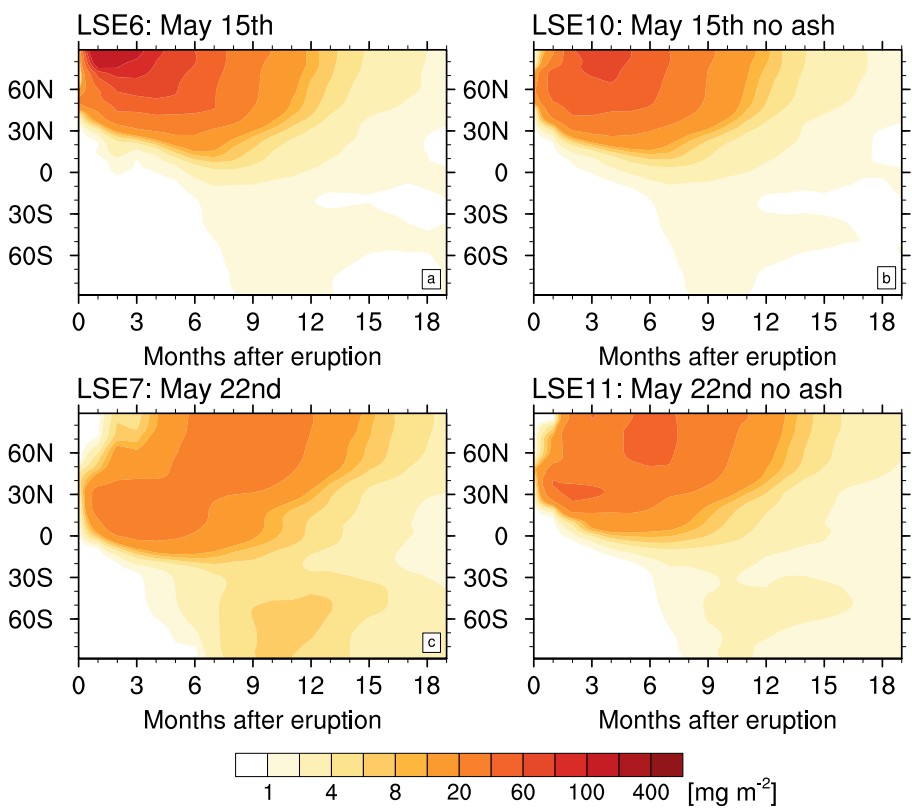

**Figure 7.** Hovmøller diagram of zonally averaged sulfate burden for an assumed LSE-like eruption over a period of 1.5 years after the first eruption phase. Eruption at May 15th (a, b) May 22nd (c, d) with injection of fine ash (a, c) and, without the injection of ash (b, c).

of dynamical conditions, we show results of LSE-like simulations, on May 15th (LSE6) and May 22nd (LSE7) respectively (Figure 7). Both have a clear north-west component of the wind, with LSE6 oriented more northward in the vicinity of an anti-cyclone (Figure 1). The distribution of sulfate is very different in both simulations (Figures 7 and 8). The slightly stronger northward transport in May at the edge of the clockwise rotating pressure system over Scandinavia in LSE6 results in a volcanic cloud mostly located between 45°N and 60°N, compared to 30°N to 50°N in LSE7 (Figure 8). This minor difference

in transport in May results, eventually, in very different sulfate burden patterns in June. Later in time, the burden maximum in LSE6 is located at the pole, but in the sub-tropics in LSE7 (Figure 7). This example underlines the importance of the specific wind pattern during the eruption influencing downstream climate impacts.

### 3.2.3 Radiative forcing

The radiative forcing at TOA of sulfate aerosols, is, in general, negative with regional values below -2.5 W m$^{-2}$ in the NH for

roughly one year after the eruption (Figure 9). During polar nights the additional absorption of near infrared and LW radiation





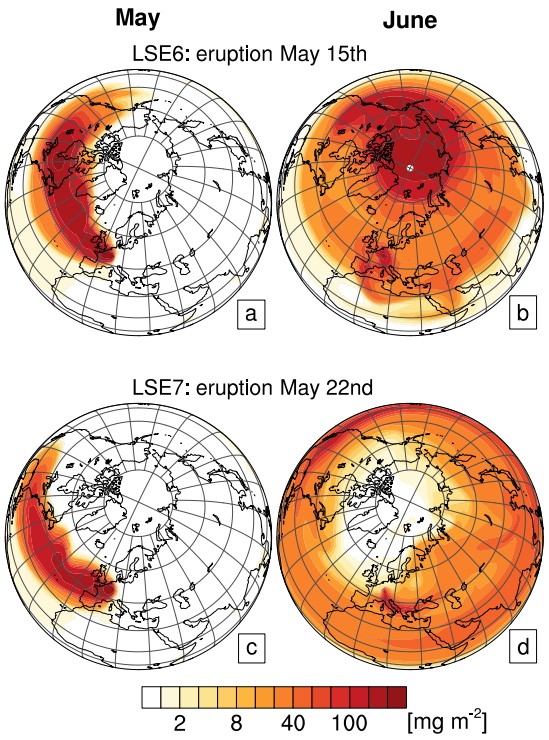

**Figure 8.** Monthly mean sulfur burden ($SO_2$ plus sulfate) shortly after the eruption (May, left and June, right) for the scenarios LSE6 (top) and LSE7 (bottom) with the same eruption rate but eruption at May 15th and May 22nd. Plotted values are 1, 2, 4, 8, 10, 20 $\mathrm{mg\,m^{-2}}$ etc.

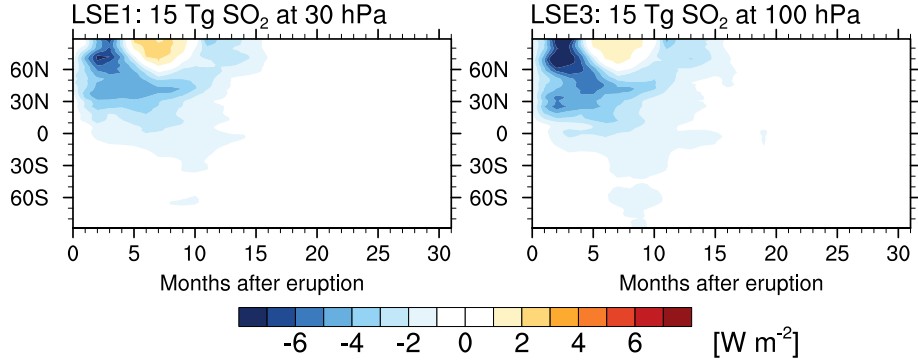

**Figure 9.** Hovmøller diagram of zonally averaged radiative forcing (all sky, top of atmosphere) of sulfate aerosols of simulation LSE1(left) and LSE3 (right). Plotted values are 0.1, 0.25, 0.5, 0.75, 1, 2.5, 5 $\mathrm{W\,m^{-2}}$.





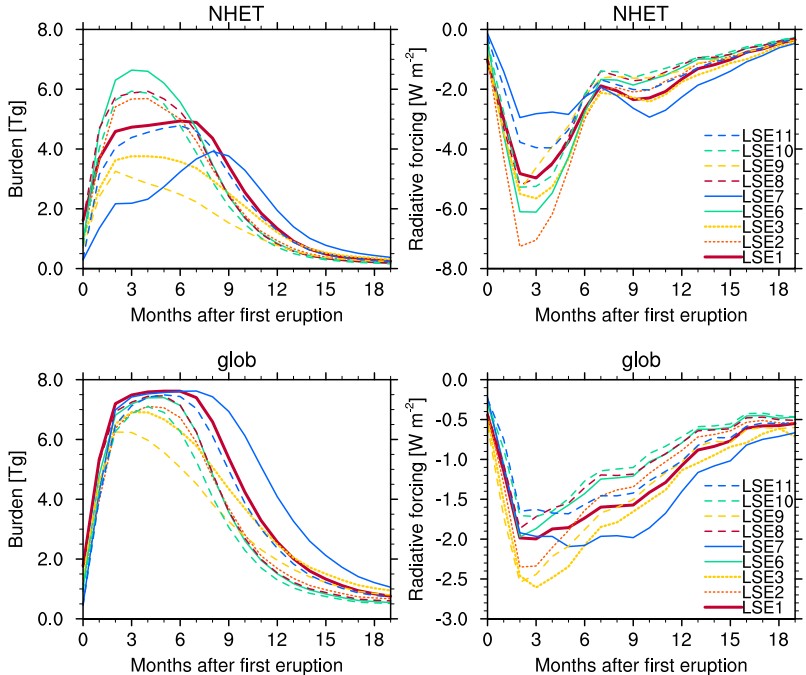

**Figure 10.** Hovmøller diagrams of sulfate burden (left) and radiative forcing (right, all sky, TOA) of sulfate aerosols. Top: averaged over the northern hemisphere extra-tropics (30° to 90°N). Bottom: Global average. Solid lines show the simulations with an initial injection of 15 Tg SO2 at different eruption days (LSE1, LSE6, LSE7), dotted lines at different injection altitudes (LSE2, LS3) and dashed lines simulations without the injection of fine ash (LSE8, LS9, LSE10).

due the volcanic aerosol lead to positive forcing anomalies around $2 \ \mathrm{Wm}^{-2}$ at high latitudes. The lower injection altitudes in LSE3 results in a slightly different anomaly pattern compared to LSE1. The negative forcing peak is slightly stronger in LSE3 and also further extended to the south. In both cases substantial negative forcing anomalies last until the end of the second summer after the eruptions not only in the NH but also the SH tropics and subtropics.

Compared to Graf and Timmreck (2001) the peak radiative forcing anomalies are smaller in our studies but reaches further into NH mid- and low-latitudes than previously simulated, mostly because of the different transport dynamics in our simulations with ash. Graf and Timmreck (2001) used a parameterization for the effective radius based on Russell et al. (1996) for the calculation of the optical parameters. These radii with peak values of 0.55 are much smaller than in our study (Suppl. Figure A2) and scatter more efficiently which could explain the higher forcing values in their study.

We compare for all our simulations with an initial emission of 15 Tg $SO_2$ global and Northern Hemisphere extratropics (NHET, 30°N to 90°N) mean values of net TOA radiative forcing and sulfate burden (Figure 10). The global burden is rather similar between the simulations in the first six months after the eruption but the decay time differs by up to four months. In sum, the higher the injection altitude the stronger the global burden maximum (LSE1 to LSE3, reddish curves). The shortest sulfate





lifetime shows LSE6, injection at May 15th, and LSE2. Both simulations show a stronger poleward transport, while LSE7
shows the longest lifetime with a strong equatorward component of the transport (Figures 4 and 7). This pattern is not fully
mirrored in the radiative forcing. Scattering of solar radiation by sulfate aerosols depends on particle size; smaller particles
scatter more strongly. The globally averaged radiative forcing is aggravated with decreasing injection altitude because particles
injected into an altitude of 100 hPa stay smaller compared to an injection into 50 hPa, a result that is in line with Toohey et al.
(2019) and Marshall et al. (2019). The smaller particle size results in stronger global forcing of LSE2 and LSE3. The ensemble
mean of the three simulations with an injection of 15 Tg S (Figure A5 with fine ash (LSE1, LSE6, LSE7) shows a higher
burden, longer lifetime and, stronger forcing in the global average compared to the ensemble without fine ash (LSE8, LSE10,
LSE11).

Interestingly, our NHET results are only partly in line with the globally averaged data. This difference is mainly caused
by transport dynamics. Simulations with a strong poleward component of transport (LSE2, LSE6) differ only slightly in their
NHET burden, while the NHET burden evolves differently for LSE7 characterised by an initial explosive eruption in late May.
Figure 7 shows the stronger southward transport of LSE7 with the main aerosols located south of 40°N until the 5th month
after the eruption. Consequently also NHET burden and radiative forcing remain smaller than in the other simulations as large
amounts of the burden did not add to the NHET values. The strongest negative radiative forcing in NHET is simulated in
LSE2 and LSE6, where most of the aerosols stay in NHET. Opposite, in LSE7 the regional impact in NHET is comparable
small but the simulation shows a strong, long lasting decrease of the global radiative forcing. In the simulations without ash
(LSE8 to LSE11), the burden and global radiative forcing of NHET are stronger in the first 6 months after the eruption (see
also ensemble mean in Figure A5), as more aerosols stay in NHET than in the corresponding simulations with ash. We discuss
details in the Discussion (Section 3.3) when comparing our results to previous studies, which have been simulated without an
injection of fine ash.

### 3.2.4   Sulfate deposition

Deposition of sulfate occurs mostly by wet deposition in the troposphere and deposition patterns are determined by the storm
tracks and the inter-tropical convergence zone. Figure 11 shows for LSE1 the global distribution of accumulated sulfate de-
position and sedimentation over 1.5 years after the eruption (calculated from monthly mean values). As expected following
a NH mid-latitude eruption deposition, values in the Southern Hemisphere are smaller than in the Northern Hemisphere, e.g
over the southern ocean values are only half of the values over the northern Atlantic. Importantly, according to our results, it
might therefore in principle be possible to find LSE sulfate deposits in Greenlandic as well as Antarctic ice cores. For LSE
eruptions of 100, 15, and 1.5 Tg (LSE5, LSE1, LSE4) respectively, we find over central Greenland (70° to 80°N, 30° to 50°W),
deposition averaged 2.5 $\mathrm{mg\,m^{-2}}$, 0.4 $\mathrm{mg\,m^{-2}}$, and 0.09 $\mathrm{mg\,m^{-2}}$ while over Antarctica (75 to 85 S, 0E to 60E), roughly 0.45
$\mathrm{mg\,m^{-2}}$, 0.05 $\mathrm{mg\,m^{-2}}$ and, up to 0.02 $\mathrm{mg\,m^{-2}}$ would have been deposited (Fig. A6). Our study thus indicates that a large NH
mid-latitude eruption such as the Laacher See eruption could have a bipolar signature (cf. (Svensson et al., 2020)). Yet, with
values just above 0.1 $\mathrm{mg\,m^{-2}}$, deposition would have been very minor indeed. Baldini et al. (2018) ascribe a large sulfate spike
at 12,867 ka BP in the GISP2 record to the Laacher See eruption. Svensson et al. (2020) identified four large bipolar sulfate

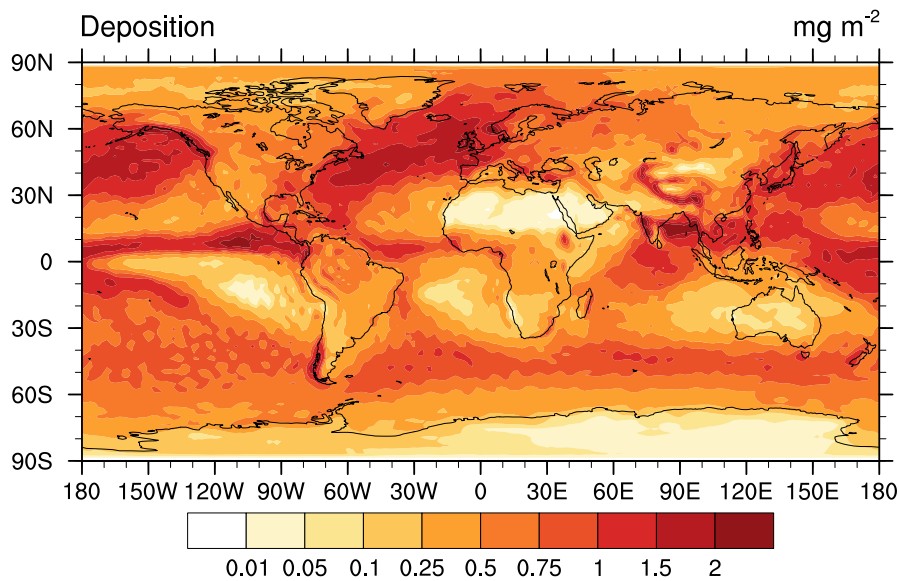

**Figure 11.** Sulfate deposition $[\mathrm{mg\ m^{-2}}]$ of the results of LSE1 accumulated over a period of 1.5 years after the first eruption.

spikes clustered around 13ka. It remains unclear whether the LSE should be associated with one of these major spikes or rather
one of the minor spikes in the decades either side - or indeed, whether we can reliably link any of these sulfate spikes with this
eruption. Increased age control on the eruption through, for instance, refined dendrochronological analyses may allow a more
confident assignment of sulfate spikes to this particular eruption.

### 3.3   Discussion

At present, only few studies exist which have investigated the climate impact of a NH mid-latitude eruption with global aerosol
models (Graf and Timmreck, 2001; Timmreck, 2001; Timmreck and Graf, 2006; Toohey et al., 2019; Marshall et al., 2019).
Comparing our results to the emulator approach of Marshall et al. (2019), who charted the radiative forcing of 100 eruptions
at different altitude, latitude and with different strength, we see broadly similar features in both studies. The effective radius of
sulfate gets smaller with lower injection altitude and at the same time increases the radiative forcing, as well as the lifetime of
the aerosols.

     When comparing our results to previous simulations of NH mid-latitude eruptions (Graf and Timmreck, 2001; Toohey et al.,
2019) we can point at small yet important differences. Both studies show a stronger transport towards high northern latitudes
than our results. Graf and Timmreck (2001) performed the first global simulation of a LSE-type eruption. Their eruption took
place in May, but they used a much larger injection area, as well as an already prescribed northeastward transport of the volcanic
cloud over the Baltic Sea. Therefore, we do not compare variables other than the radiative forcing described in Section 3.2.3.



Toohey et al. (2019) compared winter (January) and summer (July) eruptions at different NH latitudes, i.e 56°N, 36°N. The
stratospheric dynamic state is different close to the winter and summer solstice to the dynamic state in spring, as discussed
above. Hence our results are not directly comparable to those of Toohey and colleagues. In our simulation LSE7, with an
eruption day in late May, the eruption injects sulfate into a stratospheric dynamic state that is akin to summer conditions.
However, simulation LSE7 does not match well with results of the 56°N summer eruption by Toohey et al. (2019). Our
simulated sulfate transport (Figure 7) corresponds more to their pattern of a subtropical volcanic eruption at 36°N (see, Toohey
et al., 2019, Suppl. Figure 3). Both, our study and Toohey et al. (2019) use MAECHAM5-HAM, but, we also consider volcanic
ash. The additional impact of the heated ash cloud on the dynamics and flow pattern in the stratosphere causes a more southward
transport and therefore a sulfate distribution comparable to a result of a summer eruption at 36°N without the injection of
ash. Whereas the corresponding simulation without volcanic ash (LSE10) closely reflects the model results of a mid-latitude
summer eruption at 56°N by Toohey et al. (2019) with a NH high latitude maximum located between 30°N and the poles.
In general, the ensemble without ash injection shows much smaller sulfate burden between the equator and 30°N but higher
values at high latitudes shortly after the eruption (Section 3.2), resulting in lower global sulfate burden, earlier maxima and
shorter atmospheric lifetime of the aerosols (Figures 10 and A5). In line with Toohey et al. (2019) the maximum burden also
decays faster in the simulation without ash (LSE8 and LSE9) with decreasing injection altitude.

Our study strongly suggests that the injection of ash is important for the simulation of an eruption in the extra-tropics, in-
dependent of the eruption date in May. All our simulations showed a stronger transport to lower latitudes in the simulation
with ash than in the ones without. This result differs from a previous study modelling a tropical Mt. Pinatubo-like eruption by
Niemeier et al. (2009) where the impact of volcanic ash on sulfate transport was negligible. Within the tropics the rotation of
the ash cloud is less important due to the smaller Coriolis force. Our study reveals also that the development of a mesocyclonic
volcanic ash cloud depends on emission altitude and strength. Other factors which might be important are the ash size distri-
bution and gas-to-particle interactions. Our simulations neglect the latter and include only one mode of fine ash which is only
a small part of the possible spectrum of grain sizes of ejecta. Varying grain size distributions may not only alter the radiative
heating due to volcanic ash in duration and strength but also impact the onset of cloud rotation. In addition, our simulated ash
deposition shows only a fraction of the possible deposition. Models, e.g. ICON-ART (Muser et al., 2020) which consider more
ash modes and take gas-to-particle processes into account (e.g. ash coating due to sulphuric acid) may allow more detailed
studies on the impact of the rotating volcanic cloud on stratospheric dynamics and tracer transport.

MAECHAM-HAM is known for a shorter lifetime for stratospheric volcanic aerosols compared to other aerosol models
(Marshall et al., 2018; Zanchettin et al., 2016). The reason for the relative short lifetime is complex, e.g. missing OH depletion,
gravity wave parameterization, strength of meridional transport. A longer lifetime of even just a couple of months would
prolong the climate impact of the eruption, but would not lead to a dramatic climate shift. This would almost certainly require
other processes to be involved. A multi-model comparison of global aerosol models revealed that the simulated volcanic
sulfate deposition differs considerably between the models in timing, spatial pattern and magnitude due to differences in both
the transport and the formation of sulfate aerosol (Marshall et al., 2018). Deposition values should therefore be taken only
qualitatively.





For our study, we had to make assumption regarding several parameter values. One of the most critical ones is the relationship
between the ejecta of volcanic sulfur and fine ash, which we set constant to 1:10. A different fraction would certainly change
our results. The transport pattern will most likely be dominated by the amount of ash in the initial weeks after the eruptive
phase while after a month the amount of sulfate released is most important. Other factors are unknown or estimated with
high uncertainty at best. We aimed to test the sensitivity of our results to some of these uncertainties (emission rate, altitude,
meteorological conditions) but exhaustive sensitivity testing has not been possible. To arrive at a more comprehensive picture
on the impact of these parameters (e.g. injection altitude, injection duration, ash to sulfate ratio, time of the year) on the volcanic
radiative forcing and climate, emulation studies akin to that by Marshall et al. (2019) would be desirable.

## 4  Conclusions

We here report renewed attempts at modeling a large and explosive mid-latitude NH eruption akin to the cataclysmic eruption
of the Laacher See volcano around 13,000 years ago. We simulate such an eruption under volcanological and meteorological
conditions mirroring those of the Late Pleistocene eruption as documented in diverse geological, palaeoenvironmental and
archaeological archives. Our study aligns well with that of Toohey et al. (2019) in highlighting the impact potential of extra-
tropical eruptions, but complements their general model by exploring specific source parameters in a quasi-realistic scenario.
In line with previous studies, we also find that the source parameters have a substantial impact on aerosol transport as well
as downstream climatic impacts. Apart from this, we could demonstrate for the first time the importance of volcanic ash for
the burden, lifetime and radiative forcing of a large NH mid-latitude eruption. We find that heating of ash and the consequent
rotation of the ash cloud plays a crucial role in the initial transport of the fine ash and of sulfate. The additional heating of
the fine ash causes a more southward transport into areas with stronger solar irradiation, which increases the impact further.
Consequently, in this study, the sulfate burden resulting from an eruption at $50°N$ with fine ash is more comparable to a
simulation of a subtropical eruption without ash in Toohey et al. (2019). The clockwise turn of the heated air masses hinders
also the transport to higher latitudes and could, at least partly, provide a sound novel explanation for the maximum and uneven
deposition of Laacher See tephra in the eruption's medial field and the elusive tephra signal of LSE deposits in the Greenlandic
ice-cores. That said, our study does suggest that the assignment of a particular albeit almost certainly minor ice core sulfate
spike to the LSE may yet be possible, both in the Arctic and the Antarctic.

Given the dramatically different land-sea relations in the Late Pleistocene as well as differences in NH climate systems, it is
unlikely that climate models for the present day suitably capture stratospheric wind patterns for the conditions that prevailed
13,000 years ago. Our modeling study does, however, provide new insights into both the ancient eruption of the Laacher See
volcano and it provides pointers for risk assessment scenarios related to potential future volcanism in the Eifel (cf. Leder et al.
(2017)). Our initial conditions were taken to fit the modelled ash deposition to the observed lobes of the LSE. Therefore, they
depend on the specific conditions found — in our case, on a single day in early May. Such an eruption date fits well with the
date suggestions made for the ancient eruption, although it likely represents an earliest starting date. For our reference scenario
LSE1 with an injection of sulfur and ash at $30\,\mathrm{hPa}$ and $50\,\mathrm{hPa}$, we find conditions for simulating a realistic scenario in one of





three years only. At lower altitudes the wind in the stratosphere is more variable and one may find more days with wind patterns that allow an ash deposition comparable to the LSE lobe also slightly later in the year. Hence, our LSE3 simulation with an injection height of sulfur and ash at 100 hPa and 120 hPa respectively might present, under present day conditions, a more

realistic injection scenario for the LSE eruption. It also reflects the circumstances that no volcanic ash reached Greenland. The deposition pattern of fine volcanic ash also indicates that our strong emission scenario, LSE5 with an injection of 100 Tg $SO_2$ and 1000 Tg of fine ash is not likely.

Our simulations provide tantalising hints regarding the likely climatic and environmental impacts of the LSE, yet it remains difficult to asses these impacts fully from such models alone. Instead, it stands clear that the impact on climate of both the

Late Pleistocene Laacher See eruption itself as well as any future eruption scenarios have to be calculated with a fully coupled atmosphere-ocean model that, for the ancient eruption, takes account of contemporaneous land-sea relations including the fast and abrupt climate changes that occurred during the transition from the glacial to the interglacial. For future eruptions, such modelling efforts similarly need to account for the rapidly changing climatic boundary conditions of the Anthropocene.


*Code and data availability.* Primary data and scripts used in the analysis and other supplementary information that may be useful in reproducing the author's work are archived by the Max Planck Institute for Meteorology and can be obtained by contacting publications@mpimet.mpg.de. Model results will be available under cera-www.dkrz.de soon.


*Author contributions.* All authors designed the study. UN performed the simulations and the analysis, and coordinated the writing process with equal contributions of all co-authors.

*Competing interests.* The authors declare that they have no conflict of interest.



*Acknowledgements.*  We thank Christian Tegner, Anke Zernack, Anja Schmidt and Clive Oppenheimer for inspiring discussions and Traute Crüger for valuable comments on an earlier version of this paper.

This research has been supported by the Deutsche Forschungsgemeinschaft Research Unit VollImpact (FOR2820, UN CT) and by funding from the European Research Council (ERC) under the European Union's Horizon 2020 research and innovation programme (grant agreement No. 817564). The simulations were performed on the computer of the Deutsches Klima Rechenzentrum (DKRZ). Primary data and scripts
used in the analysis and other supplementary information that may be useful in reproducing the author's work are archived by the Max Planck Institute for Meteorology and can be obtained by contacting publications@mpimet.mpg.de. Model results will be available under cera-www.dkrz.de soon.





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





## Appendix A: Supplement

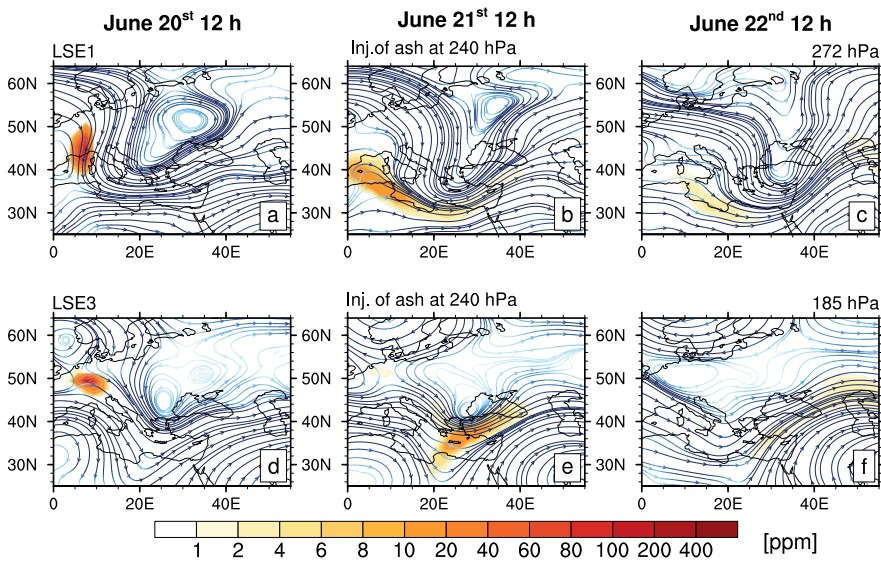

**Figure A1.** Streamlines of the zonal wind $[\mathrm{m\,s^{-1}}]$ over Europe for LSE1 at 272 hPa and LSE3 at 185 hPa at June 20th, 21st and 22nd right after the second eruption phase. The altitude shown is the level with the highest ash concentration over the three days.



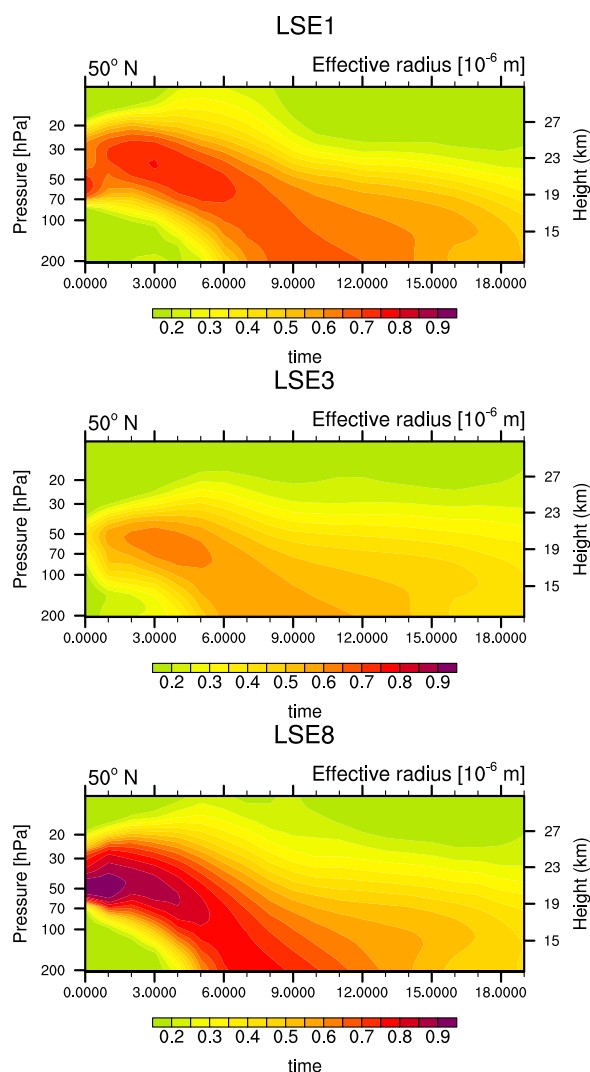

**Figure A2.** Effective radius [µm] of sulfate aerosols over time as cross section at 50°N of LSE1, LSE3 and LSE8. Injection at lower latitude (LSE3) shows smaller radii and the simulation without fine ash (LSE8) larger radii than LSE1





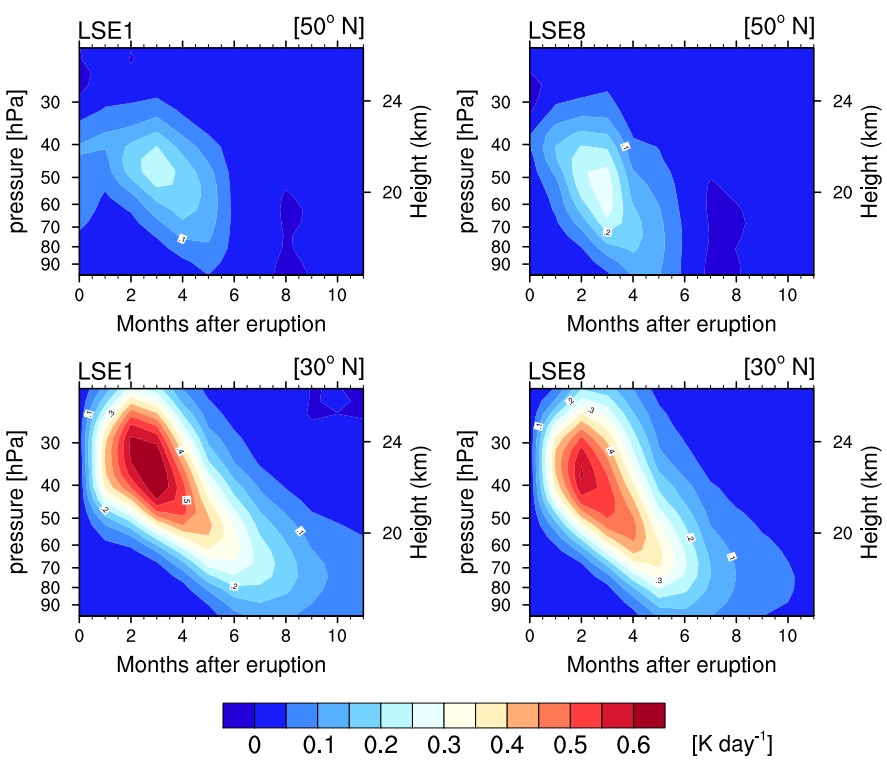

**Figure A3.** Zonal mean heating rate at $30°$(top) and $50°$N (bottom) of LSE1(left) and LSE8 (right).

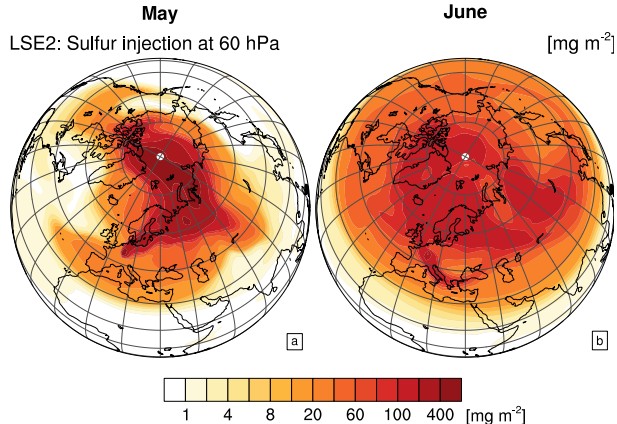

**Figure A4.** Monthly mean sulfur burden ($SO_2$ plus sulfate) shortly after the eruption (May, left and June, right) for the scenarios LSE2. Plotted values are 1, 2, 4, 8, 10, 20 $\mathrm{mg\,m^{-2}}$ etc.

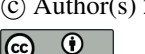

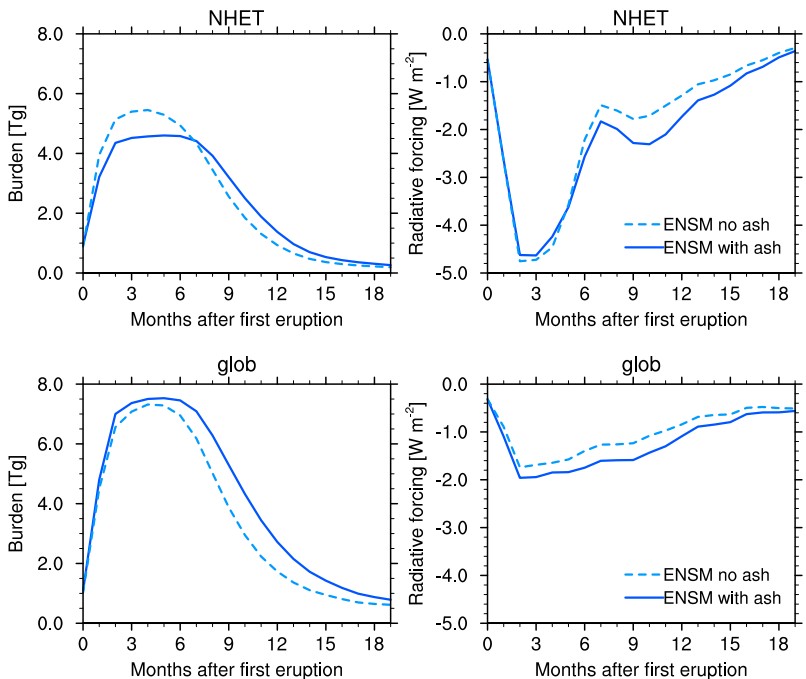

**Figure A5.** Hovmøller diagrams of the ensemble mean of sulfate burden (left) and net radiative forcing (right, all sky, top of atmosphere) of sulfate aerosols. Top: averaged over the northern hemisphere extra-tropics (30° to 90° N). Bottom: Global average. The ensemble mean was calculated of of simulations with an initial injection of 15 Tg SO2 at different eruption days, with injection of fine ash (solid, LSE1, LSE6, LSE7) and without (dashed, LSE8, LSE10, LSE11).



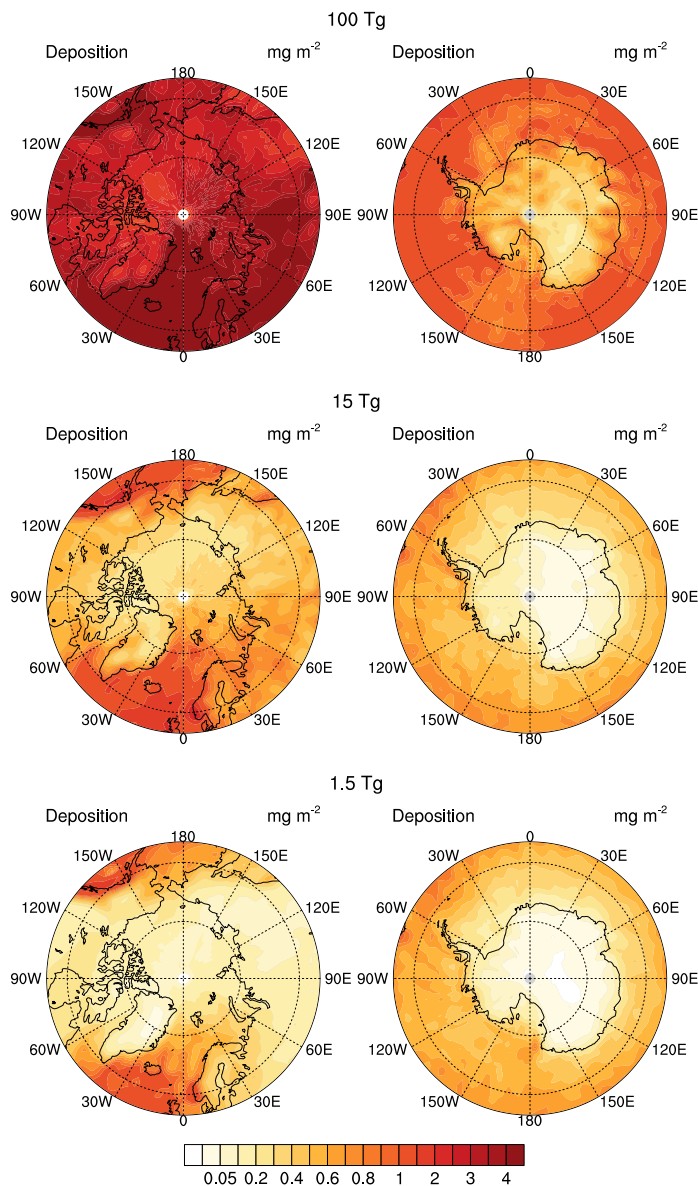

**Figure A6.** Deposition of simulations with 1.5, 15 and 100 Tg SO2. Deposition over central Greenland (70°N to 80°N, 30°W to 50°W) 2.5 mg m$^{-2}$, 0.4 mg m$^{-2}$, and 0.09 mg m$^{-2}$. Over Antarctica (75°S to 85°S , 0°to 60°E) roughly 0.45 mg m$^{-2}$, 0.05 mg, m$^{-2}$ and, up to 0.02 mg m$^{-2}$.