# Peer review of "Simulation of ash clouds after a Laacher See-type eruption"

_Climate of the Past, 2020_

## Referee Comment (RC1) · Anonymous Referee #1 · 8 Oct 2020

General comments

This study explores the potential ash clouds after a Laacher See-type eruption using a series of model simulations with different sulfur and fine ash emissions and different injection altitudes. The study picks meteorological conditions in order to best match the ash deposits and analyses the dispersion of the ash and sulfate aerosol, and the radiative impact of the sulfate aerosol. The study finds that the ash cloud rotates, which is also dependent on the altitude of the injection and that the ash also impacts the dispersion of the sulfate aerosol and consequent aerosol lifetime and radiative impact of an extra-tropical eruption.

The study is interesting, multi-disciplined, and well-structured, and I recommend publication with some minor comments.

[Figure]

Specific comments

The importance of ash for changing the sulfur distribution is clear and this could be emphasized in the conclusion as a necessity for future studies. However, I would also like to see some discussion on the importance of SO2 scavenging by ash and how this may impact your results (presumably this process is not included?) e.g., Zhu et al. 2020. Additionally, what is the role of the heating of the sulfate in changing the transport?

The simulations have two phases of the eruption, but there is little discussion of the impact of the separate phases. Can the influence of the two phases be seen in any of your results? Does it make a difference simulating the separate phases rather than the emissions all at once?

The authors state the importance of these simulations for risk assessments for future volcanism and also for understanding the social-ecological consequences of this eruption but do not give many details. Could the environmental and climatic impact (temperature, precipitation?) as predicted by these simulations be explored to support these statements?

The reconstructed ash is mentioned a lot in the introduction but not displayed until Figure 2. Could this be referred to in the text or included as a separate introductory figure, perhaps also with the reconstructed lobes?

The introduction also states that signals of this eruption in ice cores are elusive but then in section 3.2.4 mentions studies that have attributed some spikes to this eruption. This seemed a bit inconsistent.

Some of the text is difficult to read with missing words e.g. L136-142 – please check throughout.

L7 – also add the ash injection magnitudes here

L14 – 'Resulting in a stronger transport' than what?

L25 – how big is 'some' distance?

L81 – please add the length of the simulations

L136 – it is not clear to me why this might be

L146 - and also the tropospheric meteorology for deposition

Figure 1 – please add a symbol for the location of the volcano

L172-L177 – Can you justify this? Would it be better to continue each simulation to find the best meteorological condition for the second phase considering that the first phase changes the dynamics? Why are only LSE1 and LSE3 chosen for Figure A1?

L196 – can these plots show the same regional area, or can the deposits be marked on the model panels? It is hard to compare the model distribution directly with panel b.

L199 – how much is the heating?

L206-208 and throughout (e.g. L190) – It would be useful here to have the injection altitude in brackets after each simulation name, so you do not have to refer back to Table 1. Sometimes this is done but the other way around e.g. L266. Also on SI figures e.g. A2/S2.

Figure 3 – can the simulation name, ash altitude and altitude of the streamline be printed together or to the left side of the rows rather than above each separate plot as at first it was a little confusing as to what was being shown. Also, what about the results for the other simulations – these do not appear to be discussed. Are the results consistent?

L241 – extra-tropical 'northern hemisphere eruption'

L251 – Reff not explicitly shown in Marshall et al. 2019 although inferred. Perhaps better to say 'or suggest' or similar for this study

Figure 5 and others – I don't think the 'plotted values' sentence is needed, and these

values seem inconsistent with the colorbar intervals. Please check.

L288 – what do you mean by the absorption in the near infra-red is important? Why specifically ECHAM-HAM? Figure A3 comes after A4, but perhaps should be introduced earlier with the magnitude of this heating stated, for example L199.

L313 – Please add here what the radii are in this study for comparison.

Figure A2 – please check the x label and ticks

Figure 10 – Can you be more explicit with the titles - NHET burden, NHET forcing, global burden, global forcing. Not hovmoller (also Figure A5 caption)

L317 – except for LSE9? Looks different in the first 6 months.

L318 – What about LSE7, which has the largest burden? Is the meteorological condition therefore more important than injection altitude?

L319 – how is lifetime defined? It is hard to see this on the figure. What about LSE10?

L321 – also depends on the spatial evolution – is this also at play?

L325 – Tg SO2. Could you add at the end of this sentence why this is or signpost to the discussion.

L329 – I think you can be a bit more explicit here e.g. 'caused by transport dynamics and consequently the amount of aerosol that moves into the southern hemisphere'

L335 – is 'decrease' correct here? Negative radiative forcing?

L343 – is this with respect to the control/climatology?

L360 – The 100 eruptions is not strictly true - 30 eruptions were simulated, but an infinite number of eruptions can be sampled from the emulator. Perhaps remove '100'. It is a bit unclear whether you're saying the lifetime increases or decreases and the exact comparison that is being made to the emulator study – is this for an equivalent eruption at 50N and 15 Tg? This study also considered eruptions in July.

L437 – consider moving the text related to LSE3 simulations having a more realistic injection scenario to the discussion.

Technical corrections

Abbreviations are not introduced in abstract.

Appendix vs. SI figures – duplicated?

13 ka vs. 13,000 vs. 13,000 ka – please check throughout!

Table 1 – June 20 'th'. May 7th missing space for LSE3. June 20th also needed for Figure A1/S1.

L13 – add hyphen between ash and cloud?

L23 – previously 'been' suggested

L53 - 'as well as' –> 'but'

L133 - of of

L160 - without 'a'

L170 – rile –> role?

L214 – towards 'the' south or turn 'southwards'

L236 – remove 'also'

L245 – the 'sulfur' injection rate

L248 – 'at' 100 hpa

L277 – were –> where

L278 – LSE9

L285 - simulations

L306 – 'due to'

L310 – studies –> study, reaches –> reach

L317 - In sum – in general?

L322 – aggravated? Increased/magnified/larger would be better here

L334 – Opposite –> In contrast, comparable –> comparably

L371 – above –> section X

L404 - assumptions

References

Zhu, Y., Toon, O.B., Jensen, E.J. et al. Persisting volcanic ash particles impact strato-spheric SO2 lifetime and aerosol optical properties. Nat Commun 11, 4526 (2020). https://doi.org/10.1038/s41467-020-18352-5
* * *

---

## Referee Comment (RC2) · Anonymous Referee #2 · 9 Dec 2020

Review of manuscript by Ulrike Niemeier et al. for Climate of the Past journal
"Simulation of ash clouds after a Laacher See-type eruption"

This manuscript presents a very interesting analysis of interactive stratospheric
aerosol simulations of the volcanic ash and sulphuric acid aerosol cloud from
the VEI6 Laacher See eruption (Germany, 13,000 years ago,) comparing to tephra
deposits across central and northern Europe.

The interactive stratopheric aerosol modelling is novel in resolving both
components of the volcanic cloud, and in assessing how the radiative effects
from the ash, primarily via absorption of solar radiation, change the wind flow,
and the onward dispersion of the cloud in the initial weeks after the eruption.

There are some important findings in the study, and the science is certainly
within scope of the Climate of the Past journal, and the manuscript will be
publishable, but requires first some substantial improvements to better
communicate the results with also some changes of emphasis in their interpretation.

Whilst the conclusions section was well-written, I found several parts of the
manuscript were quite poorly worded, with the Abstract also needing sharpening up.
As a consequence I have listed below a large number of minor specific revisions
that need to be addressed to improve the explanation of the results before being
ready for publication.

And whereas the interactive stratospheric aerosol modelling is a clear strength
of the article, and presents a genuine advance to understand the impacts of the
ash on the sulphate cloud that causes the climate impacts, the Abstract did not
sufficiently explain the rationale to assess this, and mistakenly stated
"southerly" when southward was meant, in the clockwise rotation of the
predominantly eastward transport found in the experiments and the tephra deposits.

Another concern I have is that the Results section seems to attribute the rotation
of the cloud entirely to the heating effect from the volcanic ash. Although
clearly the volcanic cloud has important radiative effects, it's clear that even
in the control run, the flow pattern changes markedly through the weeks in May
(shown in Figures 1a to 1d), with the volcanic cloud then being transported
already with this "rotation".

Also, the word "rotation" seems non-geoscientific somehow -- it's not really a
rotation in the same way as the coriolos force. I'd argue it is more of a horizontal
wind shear effect from the change in the flow pattern (and hence the direction
the plume is transported). I suggest the authors consider changing the terminology,
although in general I guess that is consistent.  See also comment 41 that also in
other places the authors seem to mean recirculation rather than rotation.

The last sentence of the Abstract includes somewhat hyperbolic or non-scientific
wording and overall the article feels like it has not been properly checked across
the author team.

There are really a lot of places (see Main minor revisions) where the text is
poorly worded, too vague and not scientifically descriptive enough to explain the
basis of the model runs.  The study is really very interesting, and I'm sure the
authors can make the required revisions -- which are mostly minor but important
re-wordings. But because of the large number of these "Main minor revisions",
I am having to recommend major revisions in my review (but I am happy to review
the article again once the revisions have been made).

I have continued to write a long list of comments, but still have further comments
on paper. But the authors need to review the writing of the results section, and
I strongly suggest they try to improve the wording themselves before submitting
the revised version.

Main minor revisions
* * *
1) Abstract, line 6 -- the phrase "informed out experimental set-up" sounds too
technical and isn't it true to say it is the tephra deposits (from Figure 2b) that
provide the primary constraint for the model experiments?  I'm suggesting then to
be clearer and rather than "This evidence", state explicitly that the tephra is
the main constraint.  So after the more general sentence that includes also the
paleo-ecological and archaeological evidence, to make this sentence specifically
about the tephra deposits. I mean replace "This evidence has informed our
experimental setup" with "LSE tephra deposits across northern and central Europe
have provided the primary observational constraint for simulated ash deposition
as a validation test for the interactive simulations. Or similar wording to this.

2) Abstract, lines 7 and 8

This sentence in the Abstract explaining the experiments are at different injection
altitudes and emissions strengths does not need to provide the specific pressure
values for the chosen injection altitudes, and the amounts of SO2 emitted. And
doing so makes the sentence clunky and difficult to read. Suggest to delete
"(30, 60 and, 100 hPa)" and "(1.5, 15 and 100Tg SO2)" For the different injection
heights, just say "at different altitudes within the lower stratosphere" and for
the emissions amounts just say the best-estimate amounts of 20 Tg SO2 and 200Tg
of fine ash, because the lower and higher are just sensitivity runs for much larger
and much lower amounts. Nine of the 11 simulations use that 20 Tg of SO2 and 5 of
the 7 runs including ash use that 200 Tg of ash amount -- so it's only necessary
to state that value here.

Suggest to add "across two eruptive phases in May and June" and re-word the
sentence to something like:

"The interactive stratospheric aerosol model experiments are based around a central
estimate for the LSE aerosol cloud of 20 Tg of SO2 and 200 Tg of fine-ash, across
2 eruptive phases in May and June (of 10-hour and 3-hour duration), sensitivity
experiments injecting to different altitudes within the lower stratosphere.

I suggest also to add a 3rd sentence explaining about the modelling rationale to
assess the role the ultra-fine ash plays in modifying he dispersion of the volcanic
sulphur.  I mean something like:

"Additional sensitivity experiments assess how the solar-absorptive heating from
the 150 Tg of sub-micron ash emitted in the 1st eruptive phase changed the LSE
cloud's dispersion.

3) Abstract, line 9 -- The phrase "it proved difficult" is not really appropriate
in an Abstract.  And in this case, it's not really necessary to represent exactly
the meteorological conditions.  I wonder whether it's maybe more that you're
accepting it won't represent the climate conditions, but that the experiments are
designed to represent the approximate meteorological regime.

Suggest to re-word this sentence to "Whilst our simulations are based on
present-day conditions, and we do not seek to replicate the meteorolgical
[or climate?] conditions that prevailed 13,000 years ago, we consider our
experiment design to be a reasonable approximation of the transport pathways in
the mid-latitude stratosphere at this time of year."

And then continue with a new sentence providing the remainder of the sentence you
have there, but delete the word "novel" and "crucial" which might be considered
too much like hyperbole.  Suggest re-wording slightly to something more objective
such as: "The simulations show how the heating effect from the emitted fine-ash
plays an important role in the subsequent dispersion of the volcanic cloud
(both the sulphate and the ash)"

4) Abstract line 15 -- "throw new light" -- it's not possible to throw light.

I think you mean "shed light" but again this is too subjective for a journal
article. Also "awakened" seems out-of-place --  It reads like a copy and paste
from a popular science journal or proposal...  Please re-word to something more
scientific and objective, such as: "The simulations provide insight into the
impacts from the Laacher See eruption, and more generally for dispersion of
mid-latitude volcanic clouds in the stratosphere."

5) Introduction, lines 49-50 -- This sentence seems to be starting to introduce the
interactive stratospheric aerosol modelling -- and there needs to be at least
1 paragraph here (or somewhere in the Introduction) that explains about interactive
stratospheric aerosol modelling studies of volcanic ash and sulphur -- the authors'
own Niemeier et al. (2009) study is not even mentioned at all in the Abstract.

Since the paper is mainly applying an interactive stratospheric aerosol model to
simulate the volcanic aerosol cloud, the Introduction needs to have a paragraph or
2 explaining about modelling studies of similar very large magnitude volcanic
aerosol clouds, perhaps one on the interactive stratopsheric aerosol models, and
one on either modern observations from Pinatubo & El Chichon or on the impacts from
climate model studies. E.g. cite the articles by Vernier et al. (2016), and ash
modelling from mid-latitude eruptions such as 2008 Kasatochi
(e.g. Langmann et al., 2010) or more recent articles modelling the ash dispersion
from 2019 Raikoke (e.g. Muser et al., 2020).

The paragraph at the start of Results Section 3 (from lines 180 to 184) seemed to
be giving some Introduction to findings from ash modelling but that should be in
the Introduction not in the Results section.

6) Page 4, line 98 -- "For our simulations only natural sulfur emissions are taken
into account".  I'm inferring that this sentence is explaining anthropogenic sulfur
emissions are not included, but it's not clear from the wording whether it might
alternatively means there are no other natural emissions (e.g. biogenic VOCs).
And does the sentence mean DMS is included or not?  And what about passively
de-gassing volcanoes? Re-word to state more clearly what is not included that
sometimes is. "For these simulations, no anthropogenic sulphur was included, only
surface DMS emissions and tropospheric volcanic SO2 from passively degassing
volcanoes" and give the reference for the inventory used or other paper that did
the same and explained the method (e.g. as in xxx et al. 20yy).  The next sentence
says "This model" -- so are the anthropogenic emissions the same in all these
simulations even though the 2009 paper is for 1991 Pinatubo whereas this is for
13,000 years ago?  Please provide more info here to explain what is different
from those other volcanic studies and that other things are the same.  The
Niemeier and Timmreck does not emit volcanic ash -- only the Neimeier (2009)
does.  So these are claerly different models (or at least different
configurations).  The vertical resolution is here a lot higher and the version
of the atmosphere model different, right?  Please state this, as it's then good
to point out where these simulations might be more realistic.  The Niemeier et
al. (2009) model only had 39 vertical levels for example.

7) Page 4, line 104 -- "This might play a role for determining the specific day
of the eruption as discussed in Section 2.2.2".  That sentence really doesn't
make sense for a number of reasons.  Firstly, it is under-playing the
signficance of the previous sentence.  You've acknowledged that the different
Arctic sea-ice cover is likely to have changed the stratospheric circulation
and dynamics.  Even without that issue, there'd still be the different
meteorological variability between what actually happened when the eruption
occurred and the year these simulations are representing.  It's more that the
climate situation, and the timing of the disturbed meteorological situation at
vortex break-up would likely be different. Also, the current wording "might
play a role in determining the specific day of the eruption" is just the
process you've followed to get the model to represent this situation. Please
re-word this sentence to be more appropriate for the Climate of the Past
audience rather than framing it simply as a model set-up issue.

8) Page 4, line 108 -- Change "We focus here on a Laacher See-type eruption which could produce.." to something more scientific such as "The approach when setting up the model experiment, was to focus on ensuring the transport of the ash in the simulations captured the observed two-lobed pattern in the tephra deposits."  And suggest to make within that same sentence (at the end) the point about the two eruption phases. That then makes more clear to the reader the link between the timing of the 2 eruption phases and the 2 observed lobes in the tephra deposits." Suggest to delete "Hence we consider in our simulations two eruption phases", and instead say this as an extension to the previous sentence -- I mean extend "pattern in the tephra deposits" to "captured the observed pattern in the tephra deposits, the SO2 and ash emitted within 2 distinct eruption phases."  Then you can start the next sentence as follows to continue this point: "Firstly, a ten-hour-long eruption explosive phase, when the majority of the ash and sulphur were emitted, corresponding to the LLST, ash transported in the north-eastward lobe.  And secondly, a shorter and less-substantial three-hour-long phase, corresponding to the MLST-C eruption phase, when the volcanic plume deposited ash in the southward direction."

9) Page 4, line 113 -- This sentence needs to be better worded.  It's such an important part of the manuscript that sets out the basis of the modelling results and the authors should have taken much more care over this part of the manuscript.   The current wording seems out of order, and the placing of the "released into the stratosphere" straight after "two historic ash lobes" is odd -- I realise this isn't meant but the wording is clunky and since the lobes are associated with the surface, they may not of course reflect the flow pattern in the stratosphere. Also, the word "difficult" is a poor choice -- although I realise whether something is difficult or easy affects the behavioural choices people make, the wording is unscientific in this physical science context. And of course if we all made easy choices scientific progress would be slow (in my opinion).  That said, I'm simply suggesting to re-word "is difficult" to "is poorly constrained" or similar.  I suggest to put the 150 Tg ash emission in this 1st sentence, and then explain the justification of the choice afterwards, re-wording this 1st sentence as below and deleting the last 2 sentences (that information then communicated up-front in the 1st sentence).

"Only very limited information exists to determine how much fine-ash was emitted in the Laacher See model experiments, and our best-estimate fine-ash emission of 150 Tg in phase one is based on the eruption rate of 4 x 10^8 kg/s given in Textor et al. (2003), based on the 10-hour duration and approximately 1\% of the mass emitted was fine ash.

10) Page 5, lines 144-145 -- This initial sentence needs to be clearer this is referring to very large magnitude eruptions (on the scale of Pinatubo), and the word "mid-latitude" needs to be specified in the re-wording of the sentence.  . Also, suggest to add ", in the first weeks after the eruption" to spell out exactly what is meant by "initial dispersion" of the cloud.  Also, the Jones et al. (2016) study is referring to the Pinatubo cloud's dispersion within the tropical stratosphere, which is different from the mid-latitude stratosphere case considered here.  The Toohey et al. (2019) reference is fine, because that is specifically discussing mid-latitude volcanic clouds, but a different 2nd paper needs to be cited here -- I suggest replacing the Jones et al. (2016) cite instead with Marshall et al. (2019), since this JGR paper (which you have already cited in the references) explores the aerosol clouds from both tropical and mid-latitude eruptions.  Please re-word this sentence to be more specific here.

11) Page 5, line 145 -- The terms "Ash deposition pattern" and "long-range transport of volcanic ash" in this sentence are not sufficiently well defined. From the sentence after this, it sounds like you are referring to the very localised ash deposition within the proximal tephra deposits, rather than the fine-ash deposition at further distances.   But the localised tephra deposits are probably determined more by the dynamical behaviour of the plume rather

than the flow pattern in the stratosphere.  The distinction between localised
ash fallout and long-range transport of ultra-fine ash needs to be made clear
here.  Suggest extending this current sentence instead to two sentences, then
giving space to clarify there is a difference between the coarse ash particle
deposition local to the site and the stratospheric flow pattern which
determines the longer range transport.  Although I realise that the tephra
deposits have been the basis for establishing the deposition, and that is a
reasonable basis given the very long length-scale of the tephra deposits, it
still needs to be stated the role of the plume-scale processes for partially
determining the ash deposition, but that you are arguing (and I agree) that in
this case of such a very large magnitude eruption, the stratospheric flow
becomes the most important driver.

12) Page 5 lines 146-150 -- The wording of this sentence needs to be improved.
In particular, to explain better how it can be concluded this season of the
eruption, and what is meant by the environs being of a "non-analogue nature".
It is not clear to me what the authors are trying to explain here, and this
sentence is an example of the many places in the manuscript where it feels like
this manuscript has been submitted before it has been properly checked and the
text improved to a high enough standard. Perhaps they mean the local
environment is very different than today's, i.e. they meant "not analagous"
rather than "non-analogue".  But the meaning of those phrases in English are
quite different and in any case not well explained.  Suggest to discuss clearly
within the author team how best to state this difference and then be able to
justify the difference will only be second order compared to the main
circulation drivers of the volcanic plume.  Or something like that, probably
with also some re-wording of the sentences after this one.

13) Page 5 lines 151-154 -- Again this wording is poorly worded -- "that
matches an empirically known one" is not good wording and needs to be changed.
The sentence can simply state that the model experiments are not intended to
match exactly to the Laacher See tephra, but to approximate the main magnitudes
of the ash emissions in the two eruption phases,  and the change in flow
pattern that the two lobes indicated likely occurred between the two phases.
Delete "This is not possible for ancient eruptions." And I think the authors'
phrase "is a prior lead" is confused with "a priori estimate" or similar.  But
again, I think "a priori" is not the right phrase -- I'd suggest "primary
constraint" or similar.  Please re-word that sentence also accordingly.

14) Page 6, Table 1 -- Suggest to change the labels of these experiments to
make it easier for the reader to connect up the sensitivity simulations.  The
LS8 and LS9 are the no-ash equivalent runs of the LS1 and LS3 runs
(respectively) -- and it will help the reader follow what is explained if you
include already in the label that connecivity -- e.g. by labelling them
"LS1-no-ash" and "LS3-no-ash".  and LS10 and LS11 runs then labelled as
LS1-no-ash-8day abd LS1-no-ash-15day, so it's then clear immediately what those
model runs are assessing.

15) Page 6, Table 1 -- The fine-ash number-emission values given in column 3 of
Table 1 will not be meaningfult to most readers of the paper, and it's clear
from the ratio of these values to the total mass emission, that the same
emission size must be used in all 7 model experiments that emit ash. Having
those numbers alongside the mass emitted also makes the Table difficult to
scan, and since the size is the same in all runs, the number-emission value can
be given within an extra sentence in the caption, that initially states the
emissions size used.  I mean add sentence something like "In all 7 simulations
that emit fine-ash, the same emission size distribution is used, with a
geometric mean radius of xxx nm and standard deviation of y.z (particles
emitted into the accumulation insoluble mode)" -- or similar. And then "With
this emission size distribution, the 150 Tg of ash emitted in LSE1, LSE2 and
LSE3 translates to a number-emission of 2.2 x 10$^{23}$.

16) Page 6, line 160 -- More details of the control simulation need to be

provided here.  Was this a TimeSlice run with periodic boundary conditions to
repeat a particular year's conditions? Also, for how many years was the control
run spun-up prior to the analysis of which year's May meteorology provides the
required transition in the flow pattern for the two-lobed ash deposition
signature seen in the tephra deposits. An indication should also be given as to
how many years were considered to select this particular year for the main
experiments.  Was it only found in 1 year in 10 or a more common meteorological
situation than that.

17) Page 7, caption to Figure 1 -- This Figure illustrates really nicely the
change in the flow pattern that then achieves the two-lobe ash deposition
pattern seen in the tephra deposits. However, the caption here is much too
brief, and should communicate better the situation in these 1-week-separated
snapshots of the flow pattern during the transition.  The 4 panels in the
Figure are labelled a), b), c) and d), and the caption should re-iterate to the
reader the prevailing flow-direction in the region of the volcanic emission.
Suggest a sentence such as "Shown are 1-week separated snapshots of the flow
pattern through the 1st eruptive phase (LLST) with westward flow in disrupted
to be eastwards over most of Europe on May 9th (panel b) and then temporarily
Southward on May 15th (panel c), then returning to eastward flow on May 22nd
(panel d)." Or something like this.  The wording of the 2nd half of the 1st
sentence also seems out-of-order somehow, suggest to insert "on selected days
in May" before "of the control", deleting "at different days in May" at the end
of the sentence, also deleting the "at 48 hPa" at the end of the sentence, and
inserting "48 hPa" between "zonal" and "wind".  Also moving "over Europe" to be
after "Streamlines".

18) Page 7, line 180 -- I agree with the statement in the 1st sentence of the
Results section, but at the very least a citation to a paper that has shown
this is required here.  As per my Main Minor Revision 5), the Introduction
requires a paragraph explaining previous interactive ash modelling results
(including the lead author's) and I suggest here to add to that paragraph also
mention of the in-situ sampling of ash particles in the stratosphere from major
volcanic aerosol clouds: Agung (e.g. Mossop, 1964, Mossop, 1965), El Chichon
(e.g. Woods and Chuan, 1983; Gooding et al., 1983; Chuan and Woods, 1984; Rose
and Durant, 2009) and Pinatubo (e.g. Pueschel et al., 1994). The 2nd and 3rd
sentences here are also not Results and can be part of the added para in the
Introduction.

19) Page 7, line 186 -- Again, I agree with the statement in this 1st sentence
of section 3.1.1, but a reference should be cited for this, and again this
should be in the Introduction section rather than the Results.

20) Page 8, line 190 -- The wording "LSE1 shows the main deposition closest to
the Baltic Sea of all simulations" needs to be re-worded. And I suggest to
append the re-worded version of this sentence as an extension to the previous
sentence, i.e. re-word from "all simulations (Figure 2). LSE1 shows the main
deposition closest to the Baltic Sea in all simulations" with "all simulations
(Figure 2), with LSE1 showing best agreement with the LLST tephra lobe." Or
similar.

21) Page 8, lines 192-193 -- Re-word these 2 very short sentences --- you're
analysing in these 2 cases much smaller volcanic clouds (factor-10 less ash and
SO2 in LSE4) and then a much larger case, almost on the scale of Toba or so,
and this needs to be explained to the reader as you are presenting the results.
I mean to re-word to something like "The LSE4 and LSE5 cases are designed to
illustrate how the radiative effects of a very large volcanic cloud effect the
dispersion, the contrast between LSE1 and LSE4 giving the impact from the
best-estimate LSE magnitude (15Tg SO2 and 150Tg ash) to a much smaller volcanic
cloud at 1.5 Tg SO2 and 15Tg ash, then LSE5 representing a very large volcanic
cloud at 1000 Tg of ash and 100 Tg of SO2." I realise that this information is
given in the Table, but the reader needs to be reminded of the nature of these
experiments as the results are being presented.

22) Page 8, lines 213-214  -- The sentence begins "The deposition pattern of
ash..." but you need to state "model simulated" or similar so it's clear you
mean that predicted by the model experiments. More importantly, this issue of
the ash showing "a turn towards south in all cases" needs to be clarified.   As
you've shown in Figure 1, even in the control run the flow pattern is already
turning to the south for a brief period. And although I get that you're
contrasting this among the different simulations, this initial sentence
suggests it's entirely to do with the ash radiative effects. Your results do
show the effect, but you need to note initially that the flow situation already
does have a brief turn to the south in the control run.

Minor specific revisions
* * *
1) Abstract, lines 1 and 3 -- The acronym "LSE" needs to be introduced at first
use, and suggest simply to add "(LSE)" before "was one of the largest" on
Abstract line 1.

2) Abstract, line 4 -- the word "mirror" within "that mirror the empirically
known ash transport" needs to be changed as it's too precise a term.  Suggest
to replace "that mirror" with "and show can reproduce quite well".

3) Abstract, lines 5-6 -- shorten the last part of this sentence -- you've
introduced the acronym "LSE" already, and already stated the eruption occurred
in the Late Pleistocene, and I suggest to replace "Late Pleistocene eruption of
the Laacher See volcano" with "13 ka LSE". or "13 ka Laacher See eruption".

4) Abstract, line 12 -- "adds a southerly component" is too simplistic a
description, and that's an error I think -- where you say "southerly" you
actually mean "southward".  Suggest instead to reword to "acts to effectively
rotate the flow in a clockwise direction, with eastward flow changing to be
more southward."

5) Abstract, line 14 -- change "Greenlandic ice cores" to "Greenland ice
cores".

6) Introduction, line 20 -- The terms "VEI" and "M" have not been introduced,
and it's a strange choice of 1st sentence to launch straight into those indices
for the eruption.  Suggest to replace "VEI=6/M=6.2" explosive" with "very large
magnitude explosive"

7) Page 3, line 65 -- re-word "ash- and aerosol-driven" -- ash is an aerosol
particle.  I think by "aerosol-driven" you mean sulphate-driven? Please re-word
accordingly.

8) Page 3, line 79 -- insert "volcanic aerosol" before "simulations for this
study", so that it's clear immediately the model is simulating the volcanic
aerosol cloud.

9) Page 3, line 79 -- the word "GCM" has not been explained -- this acronym
could be introduced in the extra para or 2 I'm requesting in the Introduction
to provide some explanation of previous studies of ash/volcanic modeling/obs
(see main minor revision 5)

10) Page 3, line 80 -- replace "a grid size of about" with "a lat-lon grid
spacing of".

11) Page 3, lines 81-82 -- the word "evolution" seems somehow not quite right
here, suggest to replace "evolution of a volcanic cloud" with "progression of a
volcanic cloud's aerosol properties" or similar.

12) Page 3, line 82 -- Suggest to insert "an adapted version of" before "the prognostic aerosol aerosol microphysical model" -- this is a non-standard version that has been adapted to include ash, right? (e.g. with the geometric standard deviation of 1.8 and optical properties)?

13) Page 3, line 86 -- It needs to be stated here that fixed oxidant fields are used (assuming that is the case) and that the SO2 oxidation does not slow down for the large volcanic SO2 emission as the OH is used up.

14) Page 3, lines 89-90 -- insert "geometric" before "standard deviation" and "mean radius" so it's clear these are geometric mean not arithmetic mean.

15) Page 3, line 90 -- I don't understand what you mean here re: wet radius -- so does the ash take up water in the same way as in soluble modes in M7?  What hygroscopicity is assumed for the ash?  And this value of wet radius must be specific to a particular Relative Humidity or assumed volume-fraction for the water uptake in the stratosphere?  Please add a sentence to the manuscript to explain briefly how this is done in the model.

16) Page 3, line 91 -- the term "direct effect" is out-of-date -- replace "The radiative direct effect" with "aerosol-radiation radiative effect" following the terminology in AR5.

17) Page 4, line 93 -- Change "We calculate the aerosol radiative forcing" -- it's the model that calculates this as it is running (online).  And you mean the instantaneous forcing, right?  In which case suggest to re-word to "The model diagnoses the instantaneous aerosol radiative forcing each timestep, via double-call to the radiation, once with aerosol (the advancing call) and once without (an extra "diagnostic call").

18) Page 4, line 94 -- insert "both heat the stratosphere, and thereby" before "dynamically influence" and suggest to change "via temperature change" to "via circulation change" since the "heat" already communicates that the temperature will change, and the circulation change gives more insight into the subsequent effects/responses.

19) Page 4, line 107 -- Change "whom we follow here for setting the basic eruption parameter ranges" to "whose eruption chronology we follow here for setting the basic emission parameter ranges".  It's the "emission parameter" or "source parameter" rather than "eruption parameter", and its a chronology of the eruption -- that's where the term "eruption" should be used.

20) Page 4, line 114 -- with the re-wording of the 1st sentence in comment 9 of the "Main minor revisions", suggest to make this 2nd sentence continue this explanation, re-worded to instead begin "The 1% as fine ash is an estimate, with only very limited size information on the distal tephra from Laacher See (see Riede and Bazely, 2009)".

21) Page 4, line 116 -- Typo -- "Volcanic Explositivity Index" --> "Volcanic Explosivity Index".

22) Page 4, lines 118-119 -- suggest to delete the sentence "Yet, the amount of fine ash that reached the stratosphere is likely much smaller" -- that's implied in the subsequent sentence, and with the 1\% figure already cited (based on the suggested re-wording in Main Minor Revision 9), the reader will already realise this is the case.

23) Page 4, line 119 -- Replace "Pinatubo simulations (Niemeier et al., 2009).." with "When simulating the Pinatubo volcanic aerosol cloud, Niemeier et al. (2009) used the 1\% figure to determining the fine ash mass to the stratosphere, and given the large uncertainties, we consider it a reasonable approximation also for the Laacher See eruption cloud.

24) Page 5, line 156 -- Please provide a reference for the statement in this
sentence re: the change in the zonal winds in the 50-60N latitude range. The
landmark Lamb (1970) paper discusses the meteorological regimes in relation to
volcanic cloud dispersion, and although 50 years old, and focussing mainly on
the North Atlantic circulation, in relation to the British Isles, I wonder if
this or another citing a paper discussing Central Europe flow regimes should be
cited in relation to this discussion.

25) page 6, line 158 -- replace "can be more complex" with "can lead to greater
southward transport" if that is what is meant -- with also "due to more
disturbed meteorological situation" or similar.

26) Page 6, line 160 -- change "without volcanic eruption" to "without any
volcanic emission" to remind the reader the model is simulating the volcanic
aerosol cloud interactively.

27) Page 6, line 161 -- change "conditions at the LSE" to "conditions at the
time of the 13 ka LSE" or similar (to be more specific re: the particular
eruption"

28) Page 6, line 164 -- insert "the" before model initialisation" and add
afterwards "(and the volcanic ash and SO2 emission)" to communicate better it's
specifically in relation to when the eruption cloud is generated in the model.

29) Page 7, line 172 -- insert "volcanic emissions during" after "a day for
the" and replace "for the MLST-C eruption phase" with "for the 2nd eruptive
phase (MLST-C)".

30) Page 7, lines 173-174 -- replace "a date for the MLST-C phase with
transport to be" with "a date when the volcanic aerosol cloud from the 2nd
eruptive phase (MLST-C) would be transported to".

31) Page 7, lines 174-175 -- Replace "The results that best match the simulated
ash to those known empirically..." with "The meteorological situation during
the 2nd eruptive phase that gave best agreement to the MLST-C tephra
deposit..." or similar. And insert "a volcanic emission on" before "June 20th".
And then replace "This day was then used" with "This emission timing was then
chosen".

32) Page 7, line 176-177 -- Insert "as a result of the ash radiative effects"
after the open-brackets of "(Figure A1)", adding a comma before "Figure A1)".
And re-word "could only be reproduced" with "was only reproduced" (that more
accurately represents what was done, since presumably other ensemble members
approximating this situation could be chosen from a continued control run...).

33) Page 8, line 191 -- reword "lower altitudes, the maximum deposition occurs
farther to the east" with "lower altitudes, the model predicting ash deposition
much further to the east."

34) Page 8, line 192 -- Delete "is more narrow" and reword "but longer with a
more pronounced eastward spread" to "but with a longer and narrower eastward
spread".

35) Page 9, caption to Figure 2.  This panel b) is really important to see the
two tephra deposits, but the colour scale chosen is hard to distinguish (for my
eyes at least) with the grey dots, black dots and red dots.  It's also stated
that the LLST deposits are shown in brown but that looks red to me rather than
brown.  Suggest trying different colours and achieve best contrast so that it's
immediately clear to the reader which tephra deposit is which.

36) Page 9, line 215 -- Again, the reader needs to be reminded of the magnitude
of the volcanic cloud you're explaining here -- the Baines & Sparks (2005)
paper is for a super-eruption -- so if that is what you're discussing then you

should insert "For a super-eruption, " at the start of the sentence and suggest
to change "heated air" to "the heating effect from the volcanic ash" to again
be explaining more clearly to the reader the effect you're discussing.

37) Page 9, line 217 -- Re-word "a right turn" -- I think you mean "clockwise
rotation" and suggest to add "(towards the South for prevailing eastward flow)"
or similar.

38) Page 10, Figure 3 -- change "12 h" to "12 UT" in each Figure and add "12 UT
on" betwen "at" and "the 1st" in the 1st line of the caption.

39) Page 10, line 223 -- "simulation" --> "simulations".

40) Page 10, line 225 -- Add "the higher altitude volcanic cloud in" (or
similar) before "LSE1 stays closer" to better communicate the results. Also
re-word "is less strongly transported with the wind" --- do you mean the wind
speed is less?  Or is this less strongly perturbing the flow pattern in the
control? Please explain.

41) Page 10, line 226 -- Again, the wording here needs to communicate what is
different about LSE3 -- it's basically that the volcanic cloud is closer to the
tropopause -- or even at around that altitude. Change "The fast easterly
transport of of ash" to "The lower altitude volcanic cloud in LSE3, at around
the altitude of the tropopause" (or "only slightly above the tropopause" or
similar) and then continue "... is rapidly transported by the strongly eastward
wind at that altitude" or similar, and re-writing the subsequent sentence as
"with the lifting of the cloud in subsequent days....".   This "rotating cloud"
needs to be changed to "recirculating cloud" in all cases -- the word
"rotating" is not really appropriate.

References
* * *
Chuan, R. L. and Woods, D. C. (1984) "Temporal variations in characteristics of
the El Chichon stratospheric cloud" Geofisica Internacionale, vol. 23-3, pp.
335-349.

Gooding, J. L., Clanton, U. S., Gabel, E. M. and Warren, J. L. (1983) "El
Chichon volcanic ash in the stratoshpere: particle abundances and size
distribution after the 1982 eruption" Geophys. Res. Lett., vol. 10, no. 11, pp.
1033-1036.

Lamb, H. H. (1970): "Volcanic dust in the atmosphere; with a chronology and
assessment of its meteorological significance" Phil. Trans. Roy. Soc. A, vol.
266, pp. 425--533.

Langmann, B., Zaksek, K. and Hort, M. (2010): "Atmospheric distribution and
removal of volcanic ash after the eruption of Kasatochi volcano" J. Geophys.
Res., vol. 115, D00L06, doi:10.1029/2009JD013298.

Mossop, S. C. (1964): "Volcanic dust collected at an altitude of 20km" Nature,
vol. 203, pp. 824--827.

Mossop, S. C. (1965): "Stratospheric particles at 20km altitude" Geochimica et
Cosmochimica Acta, vol. 29, pp. 201--207.

Muser, O. M., Hoshyaripour, G. A., Bruckert, J., Horvath, A. et al. (2020):
"Particle aging and aerosolâ\200\223radiation interaction affect volcanic plume
dispersion: Evidence from Raikoke eruption 2019" Accepted for publication in
Atmos. Chem. Phys.: https://doi.org/10.5194/acp-2020-370

Niemeier, U., Timmreck, C., Graf, H.-F. et al. (2009): "Initial fate of fine
ash and sulfur from large volcanic eruptions" Atmos. Chem. Phys., 9, 9043-9057.

Pueschel, R. F., Russell, P. B., Allen, D. A., Ferry, G. V. et al. (1994) "Physical and optical properties of the Pinatubo volcanic aerosol: Aircraft observations with impacts and a Sun-tracking photometer" J. Geophys. Res., vol. 99, no. D6, pp. 12,915--12,922.

Rose, W. I. and Durant, A. J. (2009) "El Chichon volcano, April 4, 1982: volcanic cloud history and fine ash fallout", Nat. Hazards, vol. 51, pp. 363-374.

Vernier, J.-P., Fairlie, T. D., Deshler, T. et al. (2016) "In situ and space-based observations of the Kelud volcanic plume: The persistence of ash in the lower stratosphere" J. Geophys. Res. Atmos., vol. 121, pp. 11,104-11,118, doi:10.1002/2016JD025344.

Woods, D. C. and Chuan, R. L. (1983) "Size-specific composition of aerosols in the El Chichon volcanic cloud" Geophys. Res. Lett., vol. 10, no. 11, pp. 1041-1044.

---

## Author Comment (AC2) · 20 Jan 2021

The comment was uploaded in the form of a supplement:
https://cp.copernicus.org/preprints/cp-2020-109/cp-2020-109-AC2-supplement.pdf

———————————————

---

## Author Response (AR1)

Dear Céline,

we revised our manuscript and followed many recommendations of the reviewers. Our changes in the text were triggered by the reviewer comments, in the first part of the manuscript mostly by Reviewer 2 in the second part by questions of Reviewer 1, but also by earlier comments of Reviewer 2. They motivated us to read the text critically with a refreshed view — the long gap helped as well. Here is a short summary of our changes:

Abstract: We followed Reviewer2 and rewrote many sentences of the abstract.

Introduction: Reviewer1 recommended a new figure showing the ash lobes. Reviewer2 asked to shift a paragraph on ash from Section 3 to the Introduction, plus some words on previous work. We did both and added some words about our own paper Niemeier et al 2009.

Model and Simulations: We added details about the model and changed wording as requested by Reviewer2. We changed the names of our experiments, which was requested by both reviewers.

Results: We followed many suggestions of Reviewer2 in Section 3.1 (ash). Comments of Reviewer1, but also the critical view of Reviewer2 motivated to change and shorten the text of Section 3.2. We hope the reader can follow the read line more easily now.

Discussion: A paragraph on the differences to Niemeier et al (2009) was added, requested by Reviewer2. A discussion on SO2 scavenging on ash (Reviewer1) was added to this section and to the introduction and, we shifted a paragraph from the conclusion into the discussion (Rev1).

Figures: We added a new Figure on ash lobes (Fig1), added the location of the eruption in Fig 2 and 3, changed the order of the plots with in Fig 11 and change the order of the Figures A1, A2 and, A3 as recommended by Reviewer1. We realized that the deposition values in Figure 11 (now Figure 12) were not correct. We used a erroneous conversion factor. We corrected this error. Deposition values increased by a constant factor.

Thank you very much. Best regards,

Ulrike

**Answers to Reviewer1**

**Simulation of ash clouds after a Laacher See-type eruption**
U.Niemeier, F.Riede and, C.Timmreck

*We thank the reviewer for the kind and thoughtful comments which greatly improved the manuscript and figures. We followed the recommendations on changes in the figures, e.g. the location of the volcano in Fig 1 (now Fig2). We added a new figure (Fig 1) to the introduction showing the observed ash lobes and changed the order of the first figures in the supplement. We considered all other comments carefully and changed the text accordingly. Overall, the we changed the wording of many sentences due to the request of reviewer 2.*

*The questions and comments of the reviewers are in **bold**, answers in italic and new or changed text is marked blue.*

**General comments**

This study explores the potential ash clouds after a Laacher See-type eruption using a series of model simulations with different sulfur and fine ash emissions and different injection altitudes. The study picks meteorological conditions in order to best match the ash deposits and analyses the dispersion of the ash and sulfate aerosol, and the radiative impact of the sulfate aerosol. The study finds that the ash cloud rotates, which is also dependent on the altitude of the injection and that the ash also impacts the dispersion of the sulfate aerosol and consequent aerosol lifetime and radiative impact of an extra-tropical eruption.

The study is interesting, multi-disciplined, and well-structured, and I recommend publication with some minor comments.

**Specific comments**

**The importance of ash for changing the sulfur distribution is clear and this could be emphasized in the conclusion as a necessity for future studies.**
*Thank you. We agree that this should be emphasized a bit more. We added a few sentences to the first paragraph in the conclusion:* This shows the importance of ash for the sulfate distribution after a strong extratropical volcanic eruption. Therefore, fin ash should be taken into account in future studies — a recommendation that differs from our previous results in Niemeier et al. (2009).

**However, I would also like to see some discussion on the importance of SO2 scavenging by ash and how this may impact your results (presumably this process is not included?) e.g., Zhu et al. 2020.**
*Zhu et al. (2020) showed that the reaction of ash with sulfur, SO2 uptake on ash, causes a faster decrease of the sulfur concentration in the two months after the eruption. This reaction is not included in our model. This sulfate uptake would have an impact of the sulfate concentration. The general statement of our paper that ash adds a southerly component to transport would not change. The conclusion that the sulfate lifetime increases over a simulation without ash may change.This is difficult to estimate from the paper as they show results of the first days after the eruption only. On*

*the other hand, we parameterize the depletion of OH very roughly only, a process that increases the SO2 lifetime.*

*We changed the model description to:* To simulate the evolution of a volcanic cloud HAM was adapted to a stratospheric version (Niemeier et al., 2009). The initial conversion of SO2 into H2SO4, is simulated with a simple stratospheric sulfur chemistry scheme, which is applied above the tropopause (Timmreck, 2001; Hommel et al., 2011). We prescribe reactive gases (e.g. ozone, nitrogene oxides, hydroxyl radical (OH)) and photolysis rates of OCS, H2SO4, SO2, SO3, and O3 on a monthly mean basis. Therefore, we can parameterize the depletion of OH only: reduction of OH by 90% for the first 10 days, by 50% until 30 days after the eruption. The uptake of SO2 on ash (Zhu et al., 2020) is not included in our simulations. For these simulations, only sulfur sources relevant for stratospheric background concentration were taken into account: DMS was emitted (Stier et al., 2005) and OCS concentrations are prescribed at the surface and transported within the model. Emissions of other sources and other species are set to zero. Details on the specific stratospheric setup of HAM are described in Niemeier and Timmreck (2015). Impact of the model resolution on the results are discussed in Niemeier and Schmidt (2017) and Niemeier et al. (2020).

*We partly discuss the impact of SO2 uptake on ash in our discussion (Section 3.3) already and added a short sentence on SO2 lifetime:* Observations after the eruption of El Chichon (e.g., Woods and Chuan, 1983; CHUAN and WOODS, 2013; Pueschel et al., 1994) found ash in the atmosphere that was mantled with sulphuric acid, which could be relevant for the simulated sulfate composition in the stratosphere (Zhu et al., 2020; Muser et al., 2020). Our simulations neglect this effect, resulting in a possible slight overestimation of the SO2 lifetime. .......

**Additionally, what is the role of the heating of the sulfate in changing the transport?**

*Heating of sulfate aerosols causes a similar impact as described for ash: a southerly component is added to the transport. Aquila et al. (2012) and Timmreck and Graf (2006) have shown that without aerosol heating the volcanic cloud is transported more quickly to the pole. We added a paragraph on previous work on fine ash in the introduction. Part of the paragraph are some sentences on the role of sulfate heating.*

Sulfate aerosols absorb terrestrial and near-infrared radiation. The consequent heating of the volcanic cloud enhances transport towards the equator when aerosol-radiation interaction was incorporated into the models (Timmreck and Graf, 2006; Aquila et al., 2012).

**The simulations have two phases of the eruption, but there is little discussion of the impact of the separate phases. Can the influence of the two phases be seen in any of your results? Does it make a difference simulating the separate phases rather than the emissions all at once?**

*The second eruption is smaller, one third only, and is assumed to raise only to 220 hPa. This low altitude has two consequences: the injected sulfur does not interfere with the sulfur cloud of the first eruption as this is still at higher altitude and, the lifetime of the aerosols is shorter. One can*

*see the second eruption in Figures 5 and 8 in the burden of July in form of small maxima over mid and southern Europe. We added some sentences at the end of Section 3.2.1:* The impact of the second eruption is less strong, mainly because of the lower altitude of the eruption, but also because of the smaller eruption mass. The lower altitude avoids a strong interaction with the sulfate of the first eruption. Otherwise enhanced coagulation would cause larger particles. However, the second eruption adds to the sulfate burden as can be seen in results of June in Figure 5.

**The authors state the importance of these simulations for risk assessments for future volcanism and also for understanding the social-ecological consequences of this eruption but do not give many details. Could the environmental and climatic impact (temperature, precipitation?) as predicted by these simulations be explored to support these statements?**

*Our simulations were performed with an atmosphere only model. To give more information on the climate impact, beside the radiative forcing, the simulations have to be performed with an Earth-system model. This is planed for a future part of the related project. Therefore, we have only touched this topic at the end of the Conclusion:* Our simulations provide tantalising hints regarding the likely climatic and environmental impacts of the LSE, yet it remains difficult to asses these impacts fully from general circulation models alone. Instead, it stands clear that the impact on climate of both the Late Pleistocene Laacher See eruption itself as well as any future eruption scenarios have to be calculated with a fully coupled atmosphere-ocean model that, for the ancient eruption, takes account of contemporaneous land-sea relations including the fast and abrupt climate changes that occurred during the transition from the glacial to the interglacial. For future eruptions, such modelling efforts similarly need to account for the rapidly changing climatic boundary conditions of the Anthropocene.

**The reconstructed ash is mentioned a lot in the introduction but not displayed until Figure 2. Could this be referred to in the text or included as a separate introductory figure, perhaps also with the reconstructed lobes?**

*Thank you. We agreed and added a new Figure (Fig1).*

**The introduction also states that signals of this eruption in ice cores are elusive but then in section 3.2.4 mentions studies that have attributed some spikes to this eruption. This seemed a bit inconsistent. Some of the text is difficult to read with missing words e.g.**

*We state in the introduction that a clear chemical signal or actual tephra shards from this eruption remain unclear. In section 3.2.4 we show sulfate deposition and the text is related to sulfur. We state that there are some candidates, some spices, but they have not been identified to be caused by LSE. We changed the text in section 3.2.4:* This finding may guide the identification of Laacher See eruption signals in ice-core data. Previous identification attempts were anchored in assumed dates of the eruption and most commonly looked towards major spikes around the 13ka BP mark . Baldini et al. (2018) ascribe a large sulfate spike at 12,867 BP in the GISP2 (Greenland Ice Sheet Project) ice-core record to the Laacher See eruption. In contrast, Svensson et al. (2020) point at four large bipolar sulfate spikes clustered around 13 ka BP. It remains unclear whether the prehistoric LSE should be associated with one of these major spikes, one of the minor spikes in the adjacent decades, or whether we can at all reliably link any of these sulfate spikes with this eruption.

Increased age control on the eruption through, for instance, refined dendrochronological analyses may allow a more confident assignment of sulfate spikes to this particular eruption.

**L136-142 - please check throughout.**

*We changed the text in Section 2.2.1:*

Mid- to high-latitude eruptions might not reach as high into the stratosphere given that the erupted column reaches a buoyancy level with the local environment at lower altitude. We therefore also consider two scenarios with lower injection altitudes: 60 and, 100 hPa for SO2, and 80 and, 120 hPa for fine ash, keeping the vertical offset between the sulfur and ash emission layers constant.

**L7 - also add the ash injection magnitudes here**

*Review asked for many changes in the abstract. We changed the related sentences:* Our experiments are based around a central estimate for the Laacher See aerosol cloud of 15 Tg of sulfur dioxide (SO2) and 150 Tg of fine ash, across the main eruptive phases in May and a smaller one in June with 5 Tg SO2 and 50 Tg of fine ash. Additional sensitivity experiments reflect the estimated range of uncertainty of the injection rate and altitude and, assess how the solar-absorptive heating from the fine ash emitted in the first eruptive phase changed the volcanic cloud's dispersion.

**L14 - 'Resulting in a stronger transport' than what?** *We added:* compared to cases without injection of fine ash

**L25 - how big is 'some' distance?**

*Hundreds of kilometers. We added:* , e.g. in southern Scandinavia

**L81 - please add the length of the simulations**

*We added to Section 2.2.1:* The simulation were started from a control simulation and lasted for 1.5 years after the eruption.

**L136 - it is not clear to me why this might be**

*Within mid-latitudes the tropopause and pressure levels are at lower altitude. Therefore one might assume that the erupted column reaches a buoyancy level with the local environment at lower altitude. We changed the sentence in Section 2.2.2 to:* The 1991 Mt. Pinatubo-type eruption was a tropical one. Mid- to high-latitude eruptions might not reach as high into the stratosphere given that the erupted column reaches a buoyancy level with the local environment at lower altitude. We therefore also consider two scenarios with lower injection altitudes: 60 and, 100 hPa for SO2, and 80 and, 120 hPa for fine ash, keeping the vertical offset between the sulfur and ash emission layers constant.

**L146 - and also the tropospheric meteorology for deposition**

*Yes, but the impact is less strong. Withing the troposphere the particles are washed out quickly. Our test simulations showed a strong relation between stratospheric transport and the position of the deposited ash. We changed the text in Section 2.2.2 to:*

The initial distribution and subsequent evolution of the volcanic cloud depends on the meteorological conditions of the stratosphere at the time of the eruption (Marshall et al., 2019; Toohey et al., 2019). This is particularly pronounced in mid-latitude eruptions but holds also true for tropical eruptions

(Jones et al., 2016). Fine ash deposition patterns reflect the  transport of volcanic ash over several hundred kilometers, which is mainly determined by the meteorological situation in the lower stratosphere at the time of the eruption. Test simulations aimed at finding an appropriate injection day showed that the meteorological conditions in the troposphere were less important.

**Figure 1 - please add a symbol for the location of the volcano**

*We did, in Fig3 as well and changed the colors in Fig 3b.*

**L172-L177 - Can you justify this? Would it be better to continue each simulation to find the best meteorological condition for the second phase considering that the first phase changes the dynamics? Why are only LSE1 and LSE3 chosen for Figure A1?**

*No, we cannot really justify this decision. It was our aim to show that it is, in principle, possible to simulate the observed ash lobes. It was not our aim to show this for every case. Additionally, it was very difficult to find appropriate conditions for the right transport pattern for both eruption pulses. Therefore, we did not try to find an appropriate condition for all simulations. As stated above, the second eruption pulse is smaller and less important for the final sulfate distribution. In different parts of the text, e.g. abstract, we state now that our experiments were designed around a central experiment LSE1-30. The text changed to:* The meteorological situation that gave best agreement to the empirically known MLST-C tephra deposits were obtained for June 20th. This emission timing was chosen for all simulations despite the fact that after the firs teruptive phase LLST, the dynamic conditions changed (as a result of the ash radiative effects, Figure A2) and the deposition structure of the second explosive eruption phase was only reproduced in LSE-30.

*We decided not to show too many details and on single results. Therefore we do not show results of LSE2 (LSE60) in Fig A1 (now A2). We shortened the text in this sense as well, mostly Section 3.2.1.*

**L196 - can these plots show the same regional area, or can the deposits be marked on the model panels? It is hard to compare the model distribution directly with panel b.**

*We understand your problem. However, adding the deposits would only add another layer to the plot which would decrease the visibility of the streamlines. It was our intention to show the volcanic cloud and the rotation. This would be more difficult if we double the plotted area, as the area of interest would be very small in the plot. We decided to keep the figure unchanged. We changed the caption of Fig. 2 to:*

Note the different area in Figures e) and f) which can be seen as an extension of the area in Figure d).

**L199 - how much is the heating?**

*The heating is about 1 to 1.5 K/d in the first days after the eruption in simulation LSE2. We added a reference to Fig A3 to the sentence, as suggest later.*

**L206-208 and throughout (e.g. L190) - It would be useful here to have the injection altitude in brackets after each simulation name, so you do not have to refer back to Table 1. Sometimes this is done but the other way around e.g. L266. Also on SI figures e.g. A2/S2.**

*We changed the names of the experiments: LSE-30, LSE-60, LSE-100, LES-30-low, LSE-30-strong, LSE-30-May15 etc.*

**Figure 3 - can the simulation name, ash altitude and altitude of the streamline be printed together or to the left side of the rows rather than above each separate plot as at first it was a little confusing as to what was being shown. Also, what about there sults for the other simulations - these do not appear to be discussed. Are the results consistent?**

*We had to add this slightly strange description on top of each single plot in order to align the plots. We changed the wording slightly and hope it is less confusing now.*

**L241 - extra-tropical 'northern hemisphere eruption'**

Done

**L251 - Reff not explicitly shown in Marshall et al. 2019 although inferred. Perhaps better to say 'or suggest' or similar for this study.**

*Thank you for mentioning this, you are right. We changed the text in Section 3.2.1 to:* This is in line with previous studies: Toohey et al. (2019) show that effective radii of volcanic sulfate particles are smaller for an initial injection at 100 hPa compared to an injection at 30 hPa. Stratospheric Aerosol optical Depth and volcanic net radiative forcing results in Marshall et al. (2019) suggest a similar behaviour.

**Figure 5 and others - I don't think the 'plotted values' sentence is needed, and these values seem inconsistent with the color bar intervals. Please check.**

*We agree and changed the captions.*

**L288 - what do you mean by the absorption in the near infra-red is important? Why specifically ECHAM-HAM?**

*Absorption of radiation is treated differently in different models. In ECHAM the absorption in the near infra-red is strong, not only in terrestrial wave length. To avoid confusion, we deleted this sentence.*

**Figure A3 comes after A4, but perhaps should be introduced earlier with the magnitude of this heating stated, for example L199.**

*Thank you, we followed your suggestion.*

**L313 - Please add here what the radii are in this study for comparison.**

Larger than 0.7 $\mu$m. *We added to the text:> 0.7 $\mu m$*

**Figure A2 - please check the x label and ticks**

Done

**Figure 10 - Can you be more explicit with the titles - NHET burden, NHET forcing, global burden, global forcing. Not hovmoller (also Figure A5 caption)**

*Done. We changed the caption to:* Area average  of the ensemble mean of sulfate burden (left) and net radiative forcing....

**L317 - except for LSE9? Looks different in the first 6 months.**

*We added a* 'most' *to the text.*

**L318 - What about LSE7, which has the largest burden? Is the meteorological condition therefore more important than injection altitude?**

*The different results of LSE1, LSE6 and, LSE7 (LSE-30, LSE-30-May15, LSE-30-May22) show that the meteorological condition is very important. LSE6 and LSE7 show very different results even the initial transport in May seems to be similar (Fig.9) We discuss this differences in Section 3.2.2. Simulations LSE1, LSE6 and, LSE7 result in higher global burden than LSE2 and LSE3 with lower altitude injections. The lifetime of the volcanic cloud is shorter in LSE6, the decline of the global burden is faster, than in LSE2 and LSE3, because the particles are larger and sediment faster. This example shows that transport and aerosol microphysics interact.*

*We changed the text:*

The shortest sulfate lifetime, i.e. the fastest decay rate of all simulations with fine ash, show LSE-30-May15, and LSE-60. Both simulations show a strong poleward transport (Figures 5 and 8). LSE-30-May22 shows the latest decay of the maximum values of the global burden because of a strong equatorward component of the transport (Figure 8).

**L319 - how is lifetime defined? It is hard to see this on the figure. What about LSE10?**

*We use lifetime as a synonym for the decline of the global burden curve - earlier decay to low global burden values stands for a shorter lifetime. We added a short explanation to the text:* The shortest sulfate lifetime, i.e. the fastest decay rate of all simulations with fine ash, show LSE-30-May15 ....

**L321 - also depends on the spatial evolution - is this also at play?**

*Yes, 'the strong equartorward component of the transport', stands for spatial evolution.*

**L325 - Tg SO2. Could you add at the end of this sentence why this is or signpost to the discussion.**

*SO2: thank you for the correction. We followed your suggestion and added:* This is mainly caused be the stronger southward component in transport (see also the Discussion).

**L329 - I think you can be a bit more explicit here e.g. 'caused by transport dynamics and consequently the amount of aerosol that moves into the southern hemisphere'**

*Thank you. We agree and followed your suggestion.*

**L335 - is 'decrease' correct here? Negative radiative forcing?**

*Negative radiative forcing stands for cooling, the more negative the forcing, the stronger the cooling. Therefore, more negative forcing values describe a decreasing forcing.*

**L343 - is this with respect to the control/climatology?**

*No, the deposition values in Figure 11 are absolute deposition values. We discussed this fact and came to the conclusion that an anomaly is difficult to calculate. Circulation in a situation without volcanic eruption differs from the situation with eruption. Thus the deposition of background sulfate will not be the same with eruption than without eruption. Additionally, ice-core data include deposition of background, here natural sulfate of natural sources, as well. however, we realized that Figure 11 (now Fig.12) is not correct. We have not taken into account the monthly mean values of the original data when summing the deposition. We changes the figure accordingly.*

**L360 - The 100 eruptions is not strictly true - 30 eruptions were simulated, but an infinite number of eruptions can be sampled from the emulator. Perhaps remove '100'. It is a bit unclear whether you're saying the lifetime increases or decreases and the exact comparison that is being made to the emulator study - is this for an equivalent eruption at 50N and 15 Tg? This study also considered eruptions in July.**

*Thank you. We were so impressed by the amount of simulations that 100 can to my mind without checking the real number again. We removed the 100.*

**L437 - consider moving the text related to LSE3 simulations having a more realistic injection scenario to the discussion.**

*We moved the following sentences to the Discussion:* For our reference scenario LSE1-30 with an injection of sulfur and ash at 30 hPa and 50 hPa, we find conditions for simulating a realistic scenario in one of three years only. At lower altitudes the wind in the stratosphere is more variable and one may find more days with wind patterns that allow an ash deposition comparable to the LSE lobe also slightly later in the year. Hence, our LSE3-100 simulation with an injection height of sulfur and ash at 100 hPa and 120 hPa respectively might present, under present day conditions, a more realistic injection scenario for the LSE eruption. It also reflects the circumstances that no volcanic ash reached Greenland. The deposition pattern of fine volcanic ash also indicates that our strong emission scenario, LSE5-strong with an injection of 100 Tg SO2 and 1000 Tg of fine ash is not likely.

**Technical corrections**

**Abbreviations are not introduced in abstract.**
*We changed the abstract accordingly - no abbreviations are used.*

**Appendix vs. SI figures - duplicated?** Will be a supplement at the end.

**13 ka vs. 13,000 vs. 13,000 ka - please check throughout!** Thank you, we checked this and use 13 ka BP.

**Table 1 - June 20 'th'. May 7th missing space for LSE3. June 20th also needed for Figure A1**/S1. Done, text and figure were changed.

**L13 - add hyphen between ash and cloud?**

**L23 - previously 'been' suggested** Done

**L53 - 'as well as' -> 'but'**

**L133 - of of** Done

**L160 - without 'a'** Done

**L170 - rile -> role?** Done

**L214 - towards 'the' south or turn 'southwards'** Done

**L236 - remove 'also'** Done

**L245 - the 'sulfur' injection rate** Doe

**L248 - 'at' 100 hpa** Done

**L277 - were -> where** Done

**L278 - LSE9** Done

**L285 - simulations** Done

**L306 - 'due to'** Done

**L310 - studies -> study, reaches -> reach** Done

**L317 - In sum - in general?** Done

**L322 - aggravated? Increased/magnified/larger would be better here** We choose increased.

**L334 - Opposite -> In contrast, comparable -> comparably** Done

**L371 - above -> section X**

**L404 - assumptions** Done

**Answers to Reviewer2 on:**

**Simulation of ash clouds after a Laacher See-type eruption**
**U.Niemeier, F.Riede and, C.Timmreck**

We thank the reviewer for the very careful review and many suggestions for clarifications. We saw from the comments that some of our sentences had the capable of being misunderstood. We took your advice carefully and re-phrased many parts of the text. We agree that we should have take our own study (Niemeier et al, 2009) with more care. We included this in the introduction and discussed possible reasons for partly differing results in the Discussion. Your comments helped to improve the text and figures. We considered the recommendations carefully and made some changes in text and figures.

The questions and comments of the reviewers are in **bold**, answers in *italic* and new or changed text is marked blue.

**Answers to general comments**

**1) Abstract, line 6 – the phrase "informed out experimental set-up" sounds too technical and isn't it true to say it is the tephra deposits (from Figure 2b) that provide the primary constraint for the model experiments? I'm suggesting then to be clearer and rather than "This evidence", state explicitly that the tephra is the main constraint. So after the more general sentence that includes also the paleo-ecological and archaeological evidence, to make this sentence specifically about the tephra deposits. I mean replace "This evidence has informed our experimental setup" with "LSE tephra deposits across northern and central Europe have provided the primary observational constraint for simulated ash deposition as a validation test for the interactive simulations. Or similar wording to this.**

**2) Abstract, lines 7 and 8**
**This sentence in the Abstract explaining the experiments are at different injection altitudes and emissions strengths does not need to provide the specific pressure values for the chosen injection altitudes, and the amounts of SO2 emitted. And doing so makes the sentence clunky and difficult to read. Suggest to delete "(30, 60 and, 100 hPa)" and "(1.5, 15 and 100Tg SO2)" For the different injection heights, just say "at different altitudes within the lower stratosphere" and for the emissions amounts just say the best-estimate amounts of 20 Tg SO2 and 200Tg of fine ash, because the lower and higher are just sensitivity runs for much larger and much lower amounts. Nine of the 11 simulations use that 20 Tg of SO2 and 5 of the 7 runs including ash use that 200 Tg of ash amount – so it's only necessary to state that value here. Suggest to add "across two eruptive phases in May and June" and re-word the sentence to something like:"The interactive stratospheric aerosol model experiments are based around a central estimate for the LSE aerosol cloud of 20 Tg of SO2 and 200 Tg of fine-ash, across 2 eruptive phases in May and June (of 10-hour and 3-hour duration), sensitivity experiments injecting to different altitudes within the lower stratosphere. I suggest**

**also to add a 3rd sentence explaining about the modelling rationale to assess the role the ultra-fine ash plays in modifying the dispersion of the volcanic sulphur. I mean something like:"Additional sensitivity experiments assess how the solar-absorptive heating from the 150 Tg of sub-micron ash emitted in the 1st eruptive phase changed the LSE cloud's dispersion.**

**3) Abstract, line 9 – The phrase "it proved difficult" is not really appropriate in an Abstract. And in this case, it's not really necessary to represent exactly the meteorological conditions. I wonder whether it's maybe more that you're accepting it won't represent the climate conditions, but that the experiments are designed to represent the approximate meteorological regime.**

**Suggest to re-word this sentence to "Whilst our simulations are based on present-day conditions, and we do not seek to replicate the meteorological [or climate?] conditions that prevailed 13,000 years ago, we consider our experiment design to be a reasonable approximation of the transport pathways in the mid-latitude stratosphere at this time of year."**

**And then continue with a new sentence providing the remainder of the sentence you have there, but delete the word "novel" and "crucial" which might be considered too much like hyperbole. Suggest re-wording slightly to something more objective such as: "The simulations show how the heating effect from the emitted fine-ash plays an important role in the subsequent dispersion of the volcanic cloud (both the sulphate and the ash)"**

**4) Abstract line 15 – "throw new light" – it's not possible to throw light. I think you mean "shed light" but again this is too subjective for a journal article. Also "awakened" seems out-of-place – It reads like a copy and paste from a popular science journal or proposal... Please re-word to something more scientific and objective, such as: "The simulations provide insight into the impacts from the Laacher See eruption, and more generally for dispersion of mid-latitude volcanic clouds in the stratosphere."**

*We thank the reviewer for the very helpful suggestions. We mostly followed your advice. We were amused to realize that we might have crossed a cultural line. Germans threw light, and dictionaries tell me that the English language knows this idiom as well. To avoid confusion we changed the text. The new version of the abstract includes major and minor specific revisions 1 to 4 as well. We changed the abstract to:*

Dated to ca. 13,000 years ago, the Laacher See (East Eifel Volcanic Zone) eruption was one of the largest mid-latitude Northern Hemisphere volcanic events of the Late Pleistocene. This eruptive event not only impacted local environments and human communities but probably also effect Northern Hemistpheric climate. To better understand the impact of a Laacher See-type eruption on NH circulation and climate, we have simulated the evolution of its fine ash and sulfur cloud  with an interactive stratospheric aerosol model.  Our experiments are based around a central estimate for the Laacher See aerosol cloud of 15 Tg of sulfur dioxide (SO2) and 150 Tg of

fine ash, across the main eruptive phases in May and a smaller one in June with 5 Tg SO2 and 50 Tg of fine ash. Additional sensitivity experiments reflect the estimated range of uncertainty of the injection rate and altitude and, assess how the solar-absorptive heating from the fine ash emitted in the first eruptive phase changed the volcanic cloud's dispersion.  The chosen eruption dates were determined by the stratospheric wind fields to reflect the empirically observed ash lobes as derived from geological, palaeo-ecological and archaeological evidence linked directly to the prehistoric Laacher See eruption. Whilst our simulations are based on present-day conditions, and we do not seek to replicate the climate conditions that prevailed 13,000 years ago, we consider our experimental design to be a reasonable approximation of the transport pathways in the mid-latitude stratosphere at this time of year.  Our simulations suggest that the heating of the ash plays an important  role for the transport of ash and sulfate. Depending on the altitude of the injection, the simulated volcanic cloud begins to rotate one to three days after the eruption.  This meso-cyclone, as well as the additional radiative heating of the fine ash, then changes the dispersion of the cloud itself to be more southward compared to dispersal estimated without fine ash heating.  This ash cloud-generated southerly migration process may at least partially explain why, as yet, no Laacher See tephra has been found in Greenland ice-cores. Sulfate transport is similarly impacted by the heating of the ash, resulting in a stronger transport to low latitudes, later arrival of the volcanic cloud in the Arctic regions and, a longer lifetime compared to cases without injection of fine ash. Our study offers new insights into the dispersion of volcanic clouds in mid-latitudes and addresses a likely behaviour of the ash cloud of the Laacher See eruption that darkened European skies at the end of the Pleistocene. In turn, this study can also serve as significant input for scenarios that consider the risks associated with re-awakened volcanism in the Eifel.

**5) Introduction, lines 49-50 – This sentence seems to be starting to introduce the interactive stratospheric aerosol modelling – and there needs to be at least 1 paragraph here (or somewhere in the Introduction) that explains about interactive stratospheric aerosol modelling studies of volcanic ash and sulphur – the authors' own Niemeier et al. (2009) study is not even mentioned at all in the Abstract.**

*We agree with the reviewer and added a paragraph to the introduction and the discussion. (See below)*

**Since the paper is mainly applying an interactive stratospheric aerosol model to simulate the volcanic aerosol cloud, the Introduction needs to have a paragraph or 2 explaining about modelling studies of similar very large magnitude volcanic aerosol clouds, perhaps one on the interactive stratopsheric aerosol models, and one on either modern observations from**

**Pinatubo and El Chichon or on the impacts from climate model studies. E.g. cite the articles by Vernier et al. (2016), and ash modelling from mid-latitude eruptions such as 2008 Kasatochi (e.g. Langmann et al., 2010) or more recent articles modelling the ash dispersion from 2019 Raikoke (e.g. Muser et al., 2020).**

**The paragraph at the start of Results Section 3 (from lines 180 to 184) seemed to be giving some Introduction to findings from ash modelling but that should be in the Introduction not in the Results section.**

*We followed the reviewers advice and moved lines 180 to 184 into the introduction and added some sentences on previous publications. We agree with the reviewer that the Niemeier et al (2009) paper should have been taken a bit more serious. Therefore, we also added a new paragraph to the introduction:*

Volcanic ash has been observed with in-situ measurement techniques in the last decades (Mossop, 1964; Lamb and Sawyer, 1970; Pueschel et al., 1994; Stothers, 2001). Ash particles are relatively large and sediment quickly out of the stratosphere usually already during the first days after an eruption, although some very fine ash particles can remain in the stratosphere for longer (Vernier et al., 2016). Mossop (1964) showed that after the Mt Agung eruption only particles with diameters smaller than 0.6 $\mu$m stayed about one year or longer in the stratosphere. Ash clouds are heated by absorption of solar radiation causing an additional vertical updraft (Muser et al., 2020). This heating occurs right after the eruption, before the substantial formation of sulfate aerosols. Regionally the heating can have an impact on the transport of the volcanic cloud in the first three weeks after the eruption, but, following (Niemeier et al., 2009), the impact on global sulfate burden is small in the year after the eruption. Observations after the eruption of El Chichon (Woods and Chuan, 1983; CHUAN and WOODS, 2013; Pueschel et al., 1994) found ash in the atmosphere that was mantled with sulphuric acid. The importance of this finding for the simulated sulfate composition in the stratosphere was stated very recently by Zhu et al. (2020) and Muser et al. (2020).

*We also added some lines on the different results to Niemeier 2009 to the discussion:*

This study comes to a different result regarding the role of ash than Niemeier et al. (2009) for a simulated eruption of Mt. Katmai (58°N). They found a very small difference of global sulfur burden (1 %) between results of simulations with and without the injection of fine ash. Possible reasons for this differences can be the location, Alaska or Europe, the model resolution, T42/L37 or T63/L95, and the meteorological situation during the eruption. We stated the role of the meteorological situation before. Niemeier et al. (2009) assumed an eruption at the first of June with a clear northward flow. Figure 10 shows a smaller difference of the global burden between the simulations with the strongest northward transport (LSE6-May15 and LSE6-noash). Additional, the simulations of this study were performed with a better grid resolution. This can be an important difference when discussing the role of a meso-cyclone. We have not determined the role of resolution in detail — we will leave this for future studies. However, we assume this differences in study design as the main reason for differing results.

**6) Page 4, line 98 – "For our simulations only natural sulfur emissions are taken into ac-**

count". I'm inferring that this sentence is explaining anthropogenic sulfur emissions are not included, but it's not clear from the wording whether it might alternatively means there are no other natural emissions (e.g. biogenic VOCs). And does the sentence mean DMS is included or not? And what about passively de-gassing volcanoes? Re-word to state more clearly what is not included that sometimes is. "For these simulations, no anthropogenic sulphur was included, only surface DMS emissions and tropospheric volcanic SO2 from passively degassing volcanoes" and give the reference for the inventory used or other paper that did the same and explained the method (e.g. as in xxx et al. 20yy). The next sentence says "This model" – so are the anthropogenic emissions the same in all these simulations even though the 2009 paper is for 1991 Pinatubo whereas this is for 13,000 years ago? Please provide more info here to explain what is different from those other volcanic studies and that other things are the same. The Niemeier and Timmreck does not emit volcanic ash – only the Neimeier (2009)does. So these are claerly different models (or at least different configurations). The vertical resolution is here a lot higher and the version of the atmosphere model different, right? Please state this, as it's then good to point out where these simulations might be more realistic. The Niemeier etal. (2009) model only had 39 vertical levels for example.

*We emit sulfur species which are relevant for the stratospheric background aerosols: DMS and OCS. Sulfur from volcanic de-gasing will not be transported into the stratosphere. We use the same model in all our papers. We can change the resolution of the model in the name list. This is also true for the injection of ash and sulfur. Thus, the simulations with and without ash in this study use the same model and the same model configuration. Increasing the resolution provides a better representation of stratospheric dynamical processes e.g. the internal generation of the QBO. We changed the model description:*

We prescribe reactive gases (e.g. ozone, nitrogene oxides, hydroxyl radical (OH)) and photolysis rates of OCS, H2SO4, SO2, SO3, and O3 on a monthly mean basis. Therefore, we can parameterize the depletion of OH only: reduction of OH by 90% for the first 10 days, by 50% until 30 days after the eruption. The uptake of SO2 on ash (Zhu et al., 2020) is not included in our simulations. For these simulations, only sulfur sources relevant for stratospheric background concentration were taken into account: DMS was emitted (Stier et al., 2005) and OCS concentrations are prescribed at the surface and transported within the model. Emissions of other sources and other species are set to zero. The stratospheric setup of HAM is described in detail by Niemeier and Timmreck (2015). Earlier studies with MAECHAM5-HAM were often performed with a lower horizontal and vertical resolution, and the impact of model resolution on the results is discussed in Niemeier and Schmidt (2017) and Niemeier et al. (2020).

7) Page 4, line 104 – "This might play a role for determining the specific day of the eruption as discussed in Section 2.2.2". That sentence really doesn't make sense for a number of reasons. Firstly, it is under-playing the signficance of the previous sentence. You've acknowledged that the different Arctic sea-ice cover is likely to have changed the stratospheric

**circulation and dynamics. Even without that issue, there'd still be the different meteorological variability between what actually happened when the eruption occurred and the year these simulations are representing. It's more that the climate situation, and the timing of the disturbed meteorological situation at vortex break-up would likely be different. Also, the current wording "might play a role in determining the specific day of the eruption" is just the process you've followed to get the model to represent this situation. Please re-word this sentence to be more appropriate for the Climate of the Past audience rather than framing it simply as a model set-up issue.**

*We deleted the sentence and rephrased the paragraph:*

Although our boundary conditions are not representative for the SST during the Late Pleistocene, we assume that their impact on our results is small especially as the eruption itself almost certainly caused a strong disturbance in stratospheric flow pattern. By the same token, given that Arctic sea ice cover during the Late Pleistocene differed from today and given that stratospheric dynamics respond to sea ice conditions (Jaiser et al., 2013), it is highly likely that stratospheric conditions also differed. This might also cause the timing of the break-up of the polar vortex to differ from the year these simulations are representing. Consequently, the the specific meteorological situation that caused the observed ash lobes could have occurred later in spring than in the model world of our simulations. .

**8) Page 4, line 108 – Change "We focus here on a Laacher See-type eruption which could produce.." to something more scientific such as "The approach when setting up the model experiment, was to focus on ensuring the transport of the ash in the simulations captured the observed two-lobed pattern in the tephra deposits." And suggest to make within that same sentence (at the end) the point about the two eruption phases. That then makes more clear to the reader the link between the timing of the 2 eruption phases and the 2 observed lobesin the tephra deposits." Suggest to delete "Hence we consider in our simulations two eruption phases", and instead say this as an extension to the previous sentence – I mean extend "pattern in the tephra deposits" to"captured the observed pattern in the tephra deposits, the SO2 and ash emitted within 2 distinct eruption phases." Then you can start the next sentence asf ollows to continue this point: "Firstly, a ten-hour-long eruption explosive phase, when the majority of the ash and sulphur were emitted, corresponding to the LLST, ash transported in the north-eastward lobe. And secondly, a shorter and less-substantial three-hour-long phase, corresponding to the MLST-C eruption phase, when the volcanic plume deposited ash in the southward direction."**

*We changed the text to:*

An eruption history of the LSE has been reconstructed and described in detail by Schmincke et al. (1999)  whose eruption chronology we follow here for setting the basic emission parameter ranges. The setup of the model experiment ensures that the transport of the ash in the simulations captured the observed two-lobed

pattern in the tephra deposits within two distinct eruption phases.  Firstly, a ten hour-long strong explosive eruption phase, corresponding to the LLST, were ash is transported to the north-eastward lobe and a second less substantial three hour-long phase, corresponding to eruption phase MLST-C, depositing ash in the southward direction. The eruption is initialized over the grid box where the Laacher See is located (50.24°N, 7.16°E).

**9) Page 4, line 113 – This sentence needs to be better worded.**

*The reviewer is right this sentence is not well worded. We changed the text to:*

Only limited information exists for determining how much fine-ash has to be emitted in the model experiments. In addition, only limited particle size data.....

**10) Page 5, lines 144-145 – This initial sentence needs to be clearer this is referring to very large magnitude eruptions (on the scale of Pinatubo), and the word "mid-latitude" needs to be specified in the re-wording of the sentence. .Also, suggest to add ", in the first weeks after the eruption" to spell out exactly what is meant by "initial dispersion" of the cloud. Also, the Jones et al. (2016) study is referring to the Pinatubo cloud€™s dispersion within the tropical stratosphere, which is different from the mid-latitude stratosphere case considered here. The Toohey et al. (2019) reference is fine, because that is specifically discussing mid-latitude volcanic clouds, but a different 2nd paper needs to be cited here – I suggest replacing the Jones et al. (2016)cite instead with Marshall et al. (2019), since this JGR paper (which you have already cited in the references) explores the aerosol clouds from both tropical and mid-latitude eruptions. Please re-word this sentence to be more specific here.**

*The meteorological conditions at the altitude of the eruption determine the transport of the volcanic cloud right after the eruption. This is true for mid-latitudes, but also for the tropics, e.g. changing wind directions in the quasi-bienual oscillation. We changed the sentence:*

The distribution and subsequent evolution of the volcanic cloud depends on the meteorological conditions of the stratosphere at the time of the eruption for mid-latitude eruptions (Marshall et al., 2019; Toohey et al., 2019). This is particularlyprnounced in mid-latitude eruptions but holds also true for tropical eruptions (Jones et al., 2016).

**11) Page 5, line 145 – The terms "Ash deposition pattern" and "long-range transport of volcanic ash" in this sentence are not sufficiently well defined.From the sentence after this, it sounds like you are referring to the very localised ash deposition within the proximal tephra deposits, rather than the fine-ash deposition at further distances. But the localised tephra deposits are probably determined more by the dynamical behaviour of the plume rather than the flow pattern in the stratosphere. The distinction between localised ash fallout and long-range transport of ultra-fine ash needs to be made clear here. Suggest extending this current sentence instead to two sentences, then giving space to clarify there is a difference between the coarse ash particle deposition local to the site and the stratospheric flow pattern which**

**determines the longer range transport. Although I realise that the tephra deposits have been the basis for establishing the deposition, and that is a reasonable basis given the very long length-scale of the tephra deposits, it still needs to be stated the role of the plume-scale processes for partially determining the ash deposition, but that you are arguing (and I agree) that in this case of such a very large magnitude eruption, the stratospheric flow becomes the most important driver.**

*We agree with the reviewer that 'long-range' was not an appropriate wording. We discuss here the ash lobes a few hundred kilometers from the eruption site. There deposition occur shortly after the eruption, mostly before the rotation of the volcanic cloud forms. We changed the text:*

In our study, fine ash deposition patterns reflect the  transport of volcanic ash over some hundred kilometers,which is mainly determined by the meteorological situation in the lower stratosphere at the time of the eruption. Initial simulations aimed at finding an appropriate injection day showed that the meteorological conditions in the troposphere were less important.

**12) Page 5 lines 146-150 – The wording of this sentence needs to be improved.In particular, to explain better how it can be concluded this season of the eruption, and what is meant by the environs being of a "non-analogue nature".It is not clear to me what the authors are trying to explain here, and this sentence is an example of the many places in the manuscript where it feels like this manuscript has been submitted before it has been properly checked and the text improved to a high enough standard. Perhaps they mean the local environment is very different than today€™s, i.e. they meant "not analagous"rather than "non-analogue". But the meaning of those phrases in English are quite different and in any case not well explained. Suggest to discuss clearly within the author team how best to state this difference and then be able to justify the difference will only be second order compared to the main circulation drivers of the volcanic plume. Or something like that, probably with also some re-wording of the sentences after this one.**

*We changed the sentence. However, 'non-analogue' is a technical term in ecology:*

The palaeontological (botanical and trace-zoological) evidence preserved in the proximal LSE ash deposits offers strong indications of a late spring/early summer date of the eruption. Still, it is almost impossible to simulate an ash deposition pattern in a numerical model that  reflects exactly an empirically known one,......

**13) Page 5 lines 151-154 – Again this wording is poorly worded – "that matches an empirically known one" is not good wording and needs to be changed. The sentence can simply state that the model experiments are not intended to match exactly to the Laacher See tephra, but to approximate the main magnitudes of the ash emissions in the two eruption phases, and the change in flow pattern that the two lobes indicated likely occurred between the two phases. Delete "This is not possible for ancient eruptions." And I think the authors' phrase "is a prior lead" is confused with "a priori estimate" or similar. But again, I think "a priori" is not the right phrase – I'd suggest "primary constraint" or similar. Please re-word that sentence also accordingly.**

*We are not sure the reviewer understood our wording correctly. To prevent misunderstandings, we changed the text slightly:*

It is therefore almost impossible to simulate an ash deposition pattern in a numerical model that  reflects exactly an empirically known one, not least a deposition pattern as complicated as that of the actual LSE. For a present-day eruption, observational data could be used together with nudging (e.g. ECMWF analysis data), to push the model into a state that is similar to the weather and wind situation at the eruption day. As this nudging of meteorological variables is not possible for ancient eruptions, we used the known tephra lobe deposition as as a footprint at the surface helping us to identify possible a prior lead for the conditions in the stratosphere during the LSE in the Late Pleistocene......

**14) Page 6, Table 1 – Suggest to change the labels of these experiments to make it easier for the reader to connect up the sensitivity simulations. The LS8 and LS9 are the no-ash equivalent runs of the LS1 and LS3 runs(respectively) – and it will help the reader follow what is explained if you include already in the label that connecivity – e.g. by labelling them"LS1-no-ash" and "LS3-no-ash". and LS10 and LS11 runs then labelled as LS1-no-ash-8day abd LS1-no-ash-15day, so it's then clear immediately what those model runs are assessing.**

*The long time between handing in the manuscript and writing these answer helped us to get some distance to our text. We agree that changing the names of the experiments might be helpful. This corresponds also to remarks of reviewer 1. We changed the names to: LSE-30, LSE-60, LSE100, LSE-30-May15, LSE-30-May22, LSE-30-noash etc.*

**15) Page 6, Table 1 – The fine-ash number-emission values given in column 3 of Table 1 will not be meaningful to most readers of the paper, and it's clear from the ratio of these values to the total mass emission, that the same emission size must be used in all 7 model experiments that emit ash. Having those numbers alongside the mass emitted also makes the Table difficult to scan, and since the size is the same in all runs, the number-emission value can be given within an extra sentence in the caption, that initially states the emissions size used. I mean add sentence something like "In all 7 simulations that emit fine-ash, the same emission size distribution is used, with a geometric mean radius of xxx nm and standard deviation of y.z (particles emitted into the accumulation insoluble mode)" – or similar. And then "With this emission size distribution, the 150 Tg of ash emitted in LSE1, LSE2 and LSE3 translates to a number-emission of 2.2 x $10^{23}$.**

*We thank the reviewer for this very useful comment. We deleted this row and added to the text:*

The injected amount of particles stays in a constant ratio to the injected mass to keep the size of the injected particles the same in all simulations, e.g. 2.2 $10^{23}$ for 15 Tg (SO2) injection and 14.8$10^{23}$ for 100 Tg SO2 injection.

**16) Page 6, line 160 – More details of the control simulation need to be provided here. Was this a Time Slice run with periodic boundary conditions to repeat a particular year's conditions? Also, for how many years was the control run spun-up prior to the analysis of which year's May meteorology provides the required transition in the flow pattern for the**

**two-lobed ash deposition signature seen in the tephra deposits. An indication should also be given as to how many years were considered to select this particular year for the main experiments. Was it only found in 1 year in 10 or a more common meteorological situation than that.**

*The control simulation has been performed under the same conditions as described in Section 2.1: climatological SST values. This allows still varying meteorological conditions for different years. The text says that we looked into three different years to find an appropriate meteorological condition. We stopped to check more years thereafter. The control simulation was performed over 20 years and we used results of the years 15 to 17 to check the meteorological conditions. We added to the text:*

We performed a 20-years control simulation without any volcanic emission, with climatological SST values, monthly changing solar forcing and thus, constantly changing meteorological conditions. We checked the meteorological situation in the stratosphere in spring of three of the 20 years. In May of one year we found a situation similar to the assumed conditions at the LSE.

**17) Page 7, caption to Figure 1 – This Figure illustrates really nicely the change in the flow pattern that then achieves the two-lobe ash deposition pattern seen in the tephra deposits. However, the caption here is much too brief, and should communicate better the situation in these 1-week-separated snapshots of the flow pattern during the transition. The 4 panels in the Figure are labelled a), b), c) and d), and the caption should re-iterate to the reader the prevailing flow-direction in the region of the volcanic emission.Suggest a sentence such as "Shown are 1-week separated snapshots of the flow pattern through the 1st eruptive phase (LLST) with westward flow in disrupted to be eastwards over most of Europe on May 9th (panel b) and then temporarily Southward on May 15th (panel c), then returning to eastward flow on May 22nd(panel d)." Or something like this. The wording of the 2nd half of the 1st sentence also seems out-of-order somehow, suggest to insert "on selected days in May" before "of the control", deleting "at different days in May" at the end of the sentence, also deleting the "at 48 hPa" at the end of the sentence, and inserting "48 hPa" between "zonal" and "wind". Also moving "over Europe" to be after "Streamlines".**

*We agreed and change the text to:*

Streamlines over Europe of the undisturbed zonal wind [m s$^{-1}$] at 48 hPa. The panels show 1-week separated snapshots of the flow pattern in May with a) westward flow over central Europe on May 1st, b) eastward flow on May 7th and, north-westward flow on c) May 15th and d)Mat 22nd.

**18) Page 7, line 180 – I agree with the statement in the 1st sentence of the Results section, but at the very least a citation to a paper that has shown this is required here. As per my Main Minor Revision 5), the Introduction requires a paragraph explaining previous interactive ash modelling results(including the lead author€™s) and I suggest here to add to that paragraph also mention of the in-situ sampling of ash particles in the stratosphere from major volcanic aerosol clouds: Agung (e.g. Mossop, 1964, Mossop, 1965), El Chichon(e.g. Woods and Chuan, 1983; Gooding et al., 1983; Chuan and Woods, 1984; Rose and Durant, 2009) and**

**Pinatubo (e.g. Pueschel et al., 1994). The 2nd and 3rd sentences here are also not Results and can be part of the added para in the Introduction.**

*See answer to 5).*

**19) Page 7, line 186 – Again, I agree with the statement in this 1st sentence of section 3.1.1, but a reference should be cited for this, and again this should be in the Introduction section rather than the Results.**

*We decided to leave the text unchanged.*

**20) Page 8, line 190 – The wording "LSE1 shows the main deposition closest to the Baltic Sea of all simulations" needs to be re-worded. And I suggest to append the re-worded version of this sentence as an extension to the previous sentence, i.e. re-word from "all simulations (Figure 2). LSE1 shows the main deposition closest to the Baltic Sea in all simulations" with "all simulations(Figure 2), with LSE1 showing best agreement with the LLST tephra lobe." Or similar.**

*We followed the suggestion.*

**21) Page 8, lines 192-193 – Re-word these 2 very short sentences — you're analysing in these 2 cases much smaller volcanic clouds (factor-10 less ash and SO2 in LSE4) and then a much larger case, almost on the scale of Toba or so,and this needs to be explained to the reader as you are presenting the results.I mean to re-word to something like "The LSE4 and LSE5 cases are designed to illustrate how the radiative effects of a very large volcanic cloud effect the dispersion, the contrast between LSE1 and LSE4 giving the impact from the best-estimate LSE magnitude (15Tg SO2 and 150Tg ash) to a much smaller volcanic cloud at 1.5 Tg SO2 and 15Tg ash, then LSE5 representing a very large volcanic cloud at 1000 Tg of ash and 100 Tg of SO2." I realise that this information is given in the Table, but the reader needs to be reminded of the nature of these experiments as the results are being presented.**

*We found these information in the text. However, we clarified to:*

The pattern of deposited ash in LSE4-low is similar, but, due to a tenfold lower injected mass, the absolute value is much smaller. The opposite is the case for LSE-30-strong. The main area of deposition is similar to LSE2 and LSE3, but the spread is much greater and the ash deposits correspondingly cover a much greater area.

**22) Page 8, lines 213-214 – The sentence begins "The deposition pattern of ash..." but you need to state "model simulated" or similar so it's clear you mean that predicted by the model experiments. More importantly, this issue of the ash showing "a turn towards south in all cases" needs to be clarified. As you've shown in Figure 1, even in the control run the flow pattern is already turning to the south for a brief period. And although I get that you're contrasting this among the different simulations, this initial sentence suggests it's entirely to do with the ash radiative effects. Your results do show the effect, but you need to note initially that the flow situation already does have a brief turn to the south in the control run.**

*We agree with the reviewer, the second half of the sentence was not correct. Thank you for paying*

*attention. We changed the sentence:*

The simulated deposition pattern of ash of the LLST explosive eruption phase in May, with deposition along the Baltic Sea, differs in shape in LSE1 from LSE2 and LSE3. . This feature......

**Minor specific revisions**

**1) Abstract, lines 1 and 3 – The acronym "LSE" needs to be introduced at first use, and suggest simply to add "(LSE)" before "was one of the largest" on Abstract line 1.** Done

**2) Abstract, line 4 – the word "mirror" within "that mirror the empirically known ash transport" needs to be changed as it's too precise a term. Suggest to replace "that mirror" with "and show can reproduce quite well".**

*Change to: We reproduced...*

**3) Abstract, lines 5-6 – shorten the last part of this sentence – you've introduced the acronym "LSE" already, and already stated the eruption occurred in the Late Pleistocene, and I suggest to replace "Late Pleistocene eruption of the Laacher See volcano" with "13 ka LSE". or "13 ka Laacher See eruption".** Done. See changed abstract above.

**4) Abstract, line 12 – "adds a southerly component" is too simplistic a description, and that's an error I think – where you say "southerly" you actually mean "southward". Suggest instead to reword to "acts to effectively rotate the flow in a clockwise direction, with eastward flow changing to be more southward."** Done

**5) Abstract, line 14 – change "Greenlandic ice cores" to "Greenland icecores".** Done

**6) Introduction, line 20 – The terms "VEI" and "M" have not been introduced, and it's a strange choice of 1st sentence to launch straight into those indices for the eruption. Suggest to replace "VEI=6/M=6.2" explosive" with "very large magnitude explosive".**

*Changed to:* The very large magnitude  explosive eruption of the Laacher See volcano, Volcanic Explosivity Index 6, dated to ca. 13,000 yrs ago (13 ka before present (BP)....

**7) Page 3, line 65 – re-word "ash- and aerosol-driven" – ash is an aerosol particle. I think by "aerosol-driven" you mean sulphate-driven? Please re-word accordingly.** Done

**8) Page 3, line 79 – insert "volcanic aerosol" before "simulations for this study", so that it's clear immediately the model is simulating the volcanic aerosol cloud.** Done

**9) Page 3, line 79 – the word "GCM" has not been explained – this acronym could be introduced in the extra para or 2 I'm requesting in the Introduction to provide some explanation of previous studies of ash/volcanic modeling/obs(see main minor revision 5)**

*The reviewer is absolutely right - half of the sentence was missing. We corrected this. We also added a paragraph in the introduction on previous studies on ash (see 5).*

**10) Page 3, line 80 – replace "a grid size of about" with "a lat-lon gridspacing of".**

*ECHAM is a spectral model and does not run on a lon-lat grid. We left the sentence unchanged.*

**11) Page 3, lines 81-82 – the word "evolution" seems somehow not quite right here, suggest**

to replace "evolution of a volcanic cloud" with "progression of a volcanic cloud's aerosol properties" or similar.

*'Evolution of a volcanic cloud' is commonly used in several papers. We left the sentence unchanged.*

**12) Page 3, line 82 – Suggest to insert "an adapted version of" before "the prognostic aerosol aerosol microphysical model" – this is a non-standard version that has been adapted to include ash, right? (e.g. with the geometric standard deviation of 1.8 and optical properties)?**

*We added:* To simulate the evolution of a volcanic cloud HAM was adapted to a stratospheric version (Niemeier et al., 2009).

**13) Page 3, line 86 – It needs to be stated here that fixed oxidant fields are used (assuming that is the case) and that the SO2 oxidation does not slow down for the large volcanic SO2 emission as the OH is used up.**

*This was included together with other changes in the model description. We also state now that we parameterize OH depletion by SO2 roughly in our simulation:* We prescribe reactive gases (e.g. ozone (O3), nitrogene oxides, hydroxyl radical (OH)) and photolysis rates of Carbonyl sulfide (OCS or COS), H2SO4, SO2, SO3, and O3 on a monthly mean basis. Therefore, we can parameterize the depletion of OH only: reduction of OH by 90% for the first 10 days, and by 50% until 30 days after the eruption. The uptake of SO2 on ash (Zhu et al., 2020) is not included in our simulations. For these simulations, only sulfur sources relevant for stratospheric background concentration were taken into account: DMS was emitted (Stier et al., 2005) and OCS concentrations are prescribed at the surface and transported within the model. Emissions of other sources and other species are set to zero. The stratospheric setup of HAM is described in detail by Niemeier and Timmreck (2015). Earlier studies with MAECHAM5-HAM were often performed with a lower horizontal and vertical resolution, and the impact of model resolution on the results is discussed in Niemeier and Schmidt (2017) and Niemeier et al. (2020).

**14) Page 3, lines 89-90 – insert "geometric" before "standard deviation" and"mean radius" so it's clear these are geometric mean not arithmetic mean.** Done

**15) Page 3, line 90 – I don't understand what you mean here re: wet radius –so does the ash take up water in the same way as in soluble modes in M7? What hygroscopicity is assumed for the ash? And this value of wet radius must be specific to a particular Relative Humidity or assumed volume-fraction for the water uptake in the stratosphere? Please add a sentence to the manuscript to explain briefly how this is done in the model.**

*Ash is treated in the model as a passive tracer, but with sedimentation and deposition. Thus, no aerosol microphysical processes and no water uptake are included for ash. The radius is assumed to calculate the particle number and sedimentation velocity. We took 'wet' out.*

**16) Page 3, line 91 – the term "direct effect" is out-of-date – replace "The radiative direct effect" with "aerosol-radiation radiative effect" following the terminology in AR5.** Done

**17) Page 4, line 93 – Change "We calculate the aerosol radiative forcing" –it's the model that calculates this as it is running (online). And you mean the instantaneous forcing, right?**

**In which case suggest to re-word to "The model diagnoses the instantaneous aerosol radiative forcing each timestep, via double-call to the radiation, once with aerosol (the advancing call) and once without (an extra "diagnostic call").**

*We changed the text to:* The model diagnoses the instantaneous aerosol radiative forcing, via double-call to the radiation, once with aerosol and once with an extra diagnostic call.

**18) Page 4, line 94 – insert "both heat the stratosphere, and thereby" before"dynamically influence" and suggest to change "via temperature change" to "via circulation change" since the "heat" already communicates that the temperature will change, and the circulation change gives more insight into the subsequent effects/responses.** Done

**19) Page 4, line 107 – Change "whom we follow here for setting the basic eruption parameter ranges" to "whose eruption chronology we follow here for setting the basic emission parameter ranges". It's the "emission parameter" or"source parameter" rather than "eruption parameter", and its a chronology of the eruption – that€™s where the term "eruption" should be used.** Done

**20) Page 4, line 114 – with the re-wording of the 1st sentence in comment 9 of the "Main minor revisions", suggest to make this 2nd sentence continue this explanation, re-worded to instead begin "The 1% as fine ash is an estimate,with only very limited size information on the distal tephra from Laacher See(see Riede and Bazely, 2009)".** Done

**21) Page 4, line 116 – Typo – "Volcanic Explositivity Index" –> "Volcanic Explosivity Index".** Done

**22) Page 4, lines 118-119 – suggest to delete the sentence "Yet, the amount of fine ash that reached the stratosphere is likely much smaller" – that's implied in the subsequent sentence, and with the 1% figure already cited(based on the suggested re-wording in Main Minor Revision 9), the reader will already realise this is the case.**

*The order of our sentences is slightly different now. Therefore, we left the sentence in the text.*

**23) Page 4, line 119 – Replace "Pinatubo simulations (Niemeier et al.,2009).." with "When simulating the Pinatubo volcanic aerosol cloud, Niemeier etal. (2009) used the 1% figure to determining the fine ash mass to the stratosphere, and given the large uncertainties, we consider it a reasonable approximation also for the Laacher See eruption cloud.**

*We followed your advice.*

**24) Page 5, line 156 – Please provide a reference for the statement in this sentence re: the change in the zonal winds in the 50-60N latitude range. The landmark Lamb (1970) paper discusses the meteorological regimes in relation to volcanic cloud dispersion, and although 50 years old, and focusing mainly on the North Atlantic circulation, in relation to the British Isles, I wonder if this or another citing a paper discussing Central Europe flow regimes should be cited in relation to this discussion.**

*We could not easily get access to the full text of Lamb (1970), because of home office conditions. therefore, we cannot check if Lamb would be an appropriate citation. We decided to cite Andrews et*

*al, Middle Atmosphere Dynamics.*

**25) page 6, line 158 – replace "can be more complex" with "can lead to greater southward transport" if that is what is meant – with also "due to more disturbed meteorological situation" or similar.**

*We changed the text to:* During spring, and after the break-down of the polar vortex,  also different transport direction are possible due to more disturbed meteorological conditions with local low or high pressure systems.

**26) Page 6, line 160 – change "without volcanic eruption" to "without any volcanic emission" to remind the reader the model is simulating the volcanic aerosol cloud interactively.** Done. See also 16).

**27) Page 6, line 161 – change "conditions at the LSE" to "conditions at the time of the 13 ka LSE" or similar (to be more specific re: the particular eruption"**

*We added 'historical'*

**28) Page 6, line 164 – insert "the" before model initialisation" and add afterwards "(and the volcanic ash and SO2 emission)" to communicate better it's specifically in relation to when the eruption cloud is generated in the model.** Done

**29) Page 7, line 172 – insert "volcanic emissions during" after "a day for the" and replace "for the MLST-C eruption phase" with "for the 2nd eruptive phase (MLST-C)".** Done

**30) Page 7, lines 173-174 – replace "a date for the MLST-C phase with transport to be" with "a date when the volcanic aerosol cloud from the 2nd eruptive phase (MLST-C) would be transported to".** Done

**31) Page 7, lines 174-175 – Replace "The results that best match the simulated ash to those known empirically..." with "The meteorological situation during the 2nd eruptive phase that gave best agreement to the MLST-C tephra deposit..." or similar. And insert "a volcanic emission on" before "June 20th".And then replace "This day was then used" with "This emission timing was then chosen".** Done

**32) Page 7, line 176-177 – Insert "as a result of the ash radiative effects"after the open-brackets of "(Figure A1)", adding a comma before "Figure A1)".And re-word "could only be reproduced" with "was only reproduced" (that more accurately represents what was done, since presumably other ensemble members approximating this situation could be chosen from a continued control run...).** Done.

**33) Page 8, line 191 – reword "lower altitudes, the maximum deposition occurs farther to the east" with "lower altitudes, the model predicting ash deposition much further to the east."** Changed.

**34) Page 8, line 192 – Delete "is more narrow" and reword "but longer with a more pronounced eastward spread" to "but with a longer and narrower eastward spread".** Done.

**35) Page 9, caption to Figure 2. This panel b) is really important to see the two tephra de-**

posits, but the colour scale chosen is hard to distinguish (for my eyes at least) with the grey dots, black dots and red dots. It's also stated that the LLST deposits are shown in brown but that looks red to me rather than brown. Suggest trying different colours and achieve best contrast so that it's immediately clear to the reader which tephra deposit is which. *We tried our best and changed the colors. We added a new figure (Fig. 1) that indicates the ash lobes better.*

**36) Page 9, line 215 – Again, the reader needs to be reminded of the magnitude of the volcanic cloud you're explaining here – the Baines and Sparks (2005) paper is for a super-eruption – so if that is what you're discussing then you should insert "For a super-eruption, " at the start of the sentence and suggest to change "heated air" to "the heating effect from the volcanic ash" to again be explaining more clearly to the reader the effect you're discussing.** *Baines and Sparks (2005) discuss the size of the eruption that causes a spinning cloud. They say: "The calculations confirm that only explosive eruptions of magnitude 6.5 and above develop giant clouds a few thousand kilometres in diameter where rotational effects dominate. Mount Pinatubo in 1991 falls at the lower end of this magnitude range." We used their paper for the theoretical background. We added to our text: ...for theoretical details see explanations in (Baines.....)*

**37) Page 9, line 217 – Re-word "a right turn" – I think you mean "clockwise rotation" and suggest to add "(towards the South for prevailing eastward flow)"or similar.**
*We changed the text to:* Under the influence of the Coriolis force the horizontally expanding air turns clockwise and may even cause an anti-cyclonic spinning of the heated volcanic cloud.

**38) Page 10, Figure 3 – change "12 h" to "12 UT" in each Figure and add "12 UT on" between "at" and "the 1st" in the 1st line of the caption.** Done.

**39) Page 10, line 223 – "simulation" –> "simulations".** Done.

**40) Page 10, line 225 – Add "the higher altitude volcanic cloud in" (or similar) before "LSE1 stays closer" to better communicate the results. Also re-word "is less strongly transported with the wind" — do you mean the wind speed is less? Or is this less strongly perturbing the flow pattern in the control? Please explain.**
*We reformulated to:* The rotating ash cloud of LSE-30 stays closer to the eruptive centre as transport in the spinning cloud dominates over passive transport (compare Figures **??** a and d) and is less strongly transported with the wind. The fast passive easterly transport of ash in LSE-100 is diminished at the third day after the eruption, when the ash cloud of LSE-100 has risen, the cloud starts to rotate (May 10th) and, transport becomes dominated by the rotation of the ash cloud.

**41) Page 10, line 226 – Again, the wording here needs to communicate what is different about LSE3 – it's basically that the volcanic cloud is closer to the tropopause – or even at around that altitude. Change "The fast easterly transport of of ash" to "The lower altitude volcanic cloud in LSE3, at around the altitude of the tropopause" (or "only slightly above the tropopause" or similar) and then continue "... is rapidly transported by the strongly eastward wind at that altitude" or similar, and re-writing the subsequent sentence as"with the lifting of the cloud in subsequent days....". This "rotating cloud" needs to be changed to "recirculating**

**cloud" in all cases – the word "rotating" is not really appropriate.**

*We refer to point 40). We discussed the wording 'rotation', recirculating cloud' and, 'spinning' and decided to use rotation as done in other publications as well.*

**References**

Aquila, V., Oman, L. D., Stolarski, R. S., Colarco, P. R., and Newman, P. A.: Dispersion of the volcanic sulfate cloud from a Mount Pinatubo-like eruption, Journal of Geophysical Research: Atmospheres, 117, https://doi.org/10.1029/2011JD016968, URL `https://agupubs.onlinelibrary.wiley.com/doi/abs/10.1029/2011JD016968`, 2012.

Baldini, J. U. L., Brown, R. J., and Mawdsley, N.: Evaluating the link between the sulfur-rich Laacher See volcanic eruption and the Younger Dryas climate anomaly, Climate of the Past, 14, 969–990, https://doi.org/10.5194/cp-14-969-2018, URL `https://www.clim-past.net/14/969/2018/`, 2018.

CHUAN, R. L. and WOODS, D. C.: TEMPORAL VARIATIONS IN CHARACTERISTICS OF THE EL CHICHON STRATOSPHERIC CLOUD, GeofÃsica Internacional, 23, URL `http://revistagi.geofisica.unam.mx/index.php/RGI/article/view/1155`, 2013.

Hommel, R., Timmreck, C., and Graf, H. F.: The global middle-atmosphere aerosol model MAECHAM5-SAM2: comparison with satellite and in-situ observations, Geoscientific Model Development, 4, 809–834, https://doi.org/10.5194/gmd-4-809-2011, URL `http://www.geosci-model-dev.net/4/809/2011/`, 2011.

Jaiser, R., Dethloff, K., and Handorf, D.: Stratospheric response to Arctic sea ice retreat and associated planetary wave propagation changes, Tellus A: Dynamic Meteorology and Oceanography, 65, 19 375, https://doi.org/10.3402/tellusa.v65i0.19375, 2013.

Jones, A. C., Haywood, J. M., Jones, A., and Aquila, V.: Sensitivity of volcanic aerosol dispersion to meteorological conditions: A Pinatubo case study, Journal of Geophysical Research: Atmospheres, 121, 6892–6908, https://doi.org/10.1002/2016JD025001, URL `https://agupubs.onlinelibrary.wiley.com/doi/abs/10.1002/2016JD025001`, 2016.

Lamb, H. H. and Sawyer, J. S.: Volcanic dust in the atmosphere; with a chronology and assessment of its meteorological significance, Philosophical Transactions of the Royal Society of London. Series A, Mathematical and Physical Sciences, 266, 425–533, https://doi.org/10.1098/rsta.1970.0010, URL `https://royalsocietypublishing.org/doi/abs/10.1098/rsta.1970.0010`, 1970.

Marshall, L., Johnson, J. S., Mann, G. W., Lee, L., Dhomse, S. S., Regayre, L., Yoshioka, M., Carslaw, K. S., and Schmidt, A.: Exploring How Eruption Source Parameters Affect Volcanic

Radiative Forcing Using Statistical Emulation, Journal of Geophysical Research: Atmospheres, 124, 964–985, https://doi.org/10.1029/2018JD028675, URL `https://agupubs.onlinelibrary.wiley.com/doi/abs/10.1029/2018JD028675`, 2019.

Mossop, S.: Volcanic Dust Collected at an Altitude of 20 KM, Nature, 203, 824–827, https://doi.org/https://doi.org/10.1038/203824a0, 1964.

Muser, L. O., Hoshyaripour, G. A., Bruckert, J., Horvath, A., Malinina, E., Peglow, S., Prata, F. J., Rozanov, A., von Savigny, C., Vogel, H., and Vogel, B.: Particle Aging and Aerosol–Radiation Interaction Affect Volcanic Plume Dispersion: Evidence from Raikoke Eruption 2019, Atmospheric Chemistry and Physics Discussions, 2020, 1–27, https://doi.org/10.5194/acp-2020-370, URL `https://www.atmos-chem-phys-discuss.net/acp-2020-370/`, 2020.

Niemeier, U. and Schmidt, H.: Changing transport processes in the stratosphere by radiative heating of sulfate aerosols, Atmospheric Chemistry and Physics, 17, 14871–14886, https://doi.org/10.5194/acp-17-14871-2017, URL `https://www.atmos-chem-phys.net/17/14871/2017/`, 2017.

Niemeier, U. and Timmreck, C.: What is the limit of climate engineering by stratospheric injection of SO2?, Atmospheric Chemistry and Physics, 15, 9129–9141, https://doi.org/10.5194/acp-15-9129-2015, URL `http://www.atmos-chem-phys.net/15/9129/2015/`, 2015.

Niemeier, U., Timmreck, C., Graf, H.-F., Kinne, S., Rast, S., and Self, S.: Initial fate of fine ash and sulfur from large volcanic eruptions, Atmospheric Chemistry and Physics, 9, 9043–9057, URL `http://www.atmos-chem-phys.net/9/9043/2009/`, 2009.

Niemeier, U., Richter, J. H., and Tilmes, S.: Differing responses of the QBO to SO2 injections in two global models, Atmospheric Chemistry and Physics Discussions, 2020, 1–21, https://doi.org/10.5194/acp-2020-206, URL `https://www.atmos-chem-phys-discuss.net/acp-2020-206/`, 2020.

Pueschel, R. F., Russell, P. B., Allen, D. A., Ferry, G. V., Snetsinger, K. G., Livingston, J. M., and Verma, S.: Physical and optical properties of the Pinatubo volcanic aerosol: Aircraft observations with impactors and a Sun-tracking photometer, Journal of Geophysical Research: Atmospheres, 99, 12915–12922, https://doi.org/https://doi.org/10.1029/94JD00621, URL `https://agupubs.onlinelibrary.wiley.com/doi/abs/10.1029/94JD00621`, 1994.

Schmincke, H.-U., Park, C., and Harms, E.: Evolution and environmental impacts of the eruption of Laacher See Volcano (Germany) 12,900 a BP, Quaternary International, 61, 61 − 72, https://doi.org/https://doi.org/10.1016/S1040-6182(99)00017-8, URL `http://www.sciencedirect.com/science/article/pii/S1040618299000178`, 1999.

Stier, P., Feichter, J., Kinne, S., Kloster, S., Vignati, E., Wilson, J., Ganzeveld, L., Tegen, I., Werner, M., Balkanski, Y., Schulz, M., Boucher, O., Minikin, A., and Petzold, A.: The aerosol–climate model ECHAM5–HAM, Atmos. Chem. Phys., 5, 1125–1156, 2005.

Stothers, R. B.: Major optical depth perturbations to the stratosphere from volcanic eruptions: Stellar extinction period, 1961–1978, Journal of Geophysical Research: Atmospheres, 106, 2993–3003, https://doi.org/10.1029/2000JD900652, 2001.

Svensson, A., Dahl-Jensen, D., Steffensen, J. P., Blunier, T., Rasmussen, S. O., Vinther, B. M., Vallelonga, P., Capron, E., Gkinis, V., Cook, E., Kjær, H. A., Muscheler, R., Kipfstuhl, S., Wilhelms, F., Stocker, T. F., Fischer, H., Adolphi, F., Erhardt, T., Sigl, M., Landais, A., Parrenin, F., Buizert, C., McConnell, J. R., Severi, M., Mulvaney, R., and Bigler, M.: Bipolar volcanic synchronization of abrupt climate change in Greenland and Antarctic ice cores during the last glacial period, Climate of the Past, 16, 1565–1580, https://doi.org/10.5194/cp-16-1565-2020, URL https://cp.copernicus.org/articles/16/1565/2020/, 2020.

Timmreck, C.: Three–dimensional simulation of stratospheric background aerosol: First results of a multiannual general circulation model simulation, J. Geophys. Res., 106, 28 313–28 332, 2001.

Timmreck, C. and Graf, H.-F.: The initial dispersal and radiative forcing of a Northern Hemisphere mid–latitude super volcano: A model study, Atmos. Chem. Phys., 6, 35–49, URL http://www.atmos-chem-phys.net/6/35/2006/, 2006.

Toohey, M., Krüger, K., Schmidt, H., Timmreck, C., Sigl, M., Stoffel, M., and Wilson, R.: Disproportionately strong climate forcing from extratropical explosive volcanic eruptions., Nature Geosci, 12, 100–107, https://doi.org/10.1038/s41561-018-0286-2, 2019.

Vernier, J.-P., Fairlie, T. D., Deshler, T., Natarajan, M., Knepp, T., Foster, K., Wienhold, F. G., Bedka, K. M., Thomason, L., and Trepte, C.: In situ and space-based observations of the Kelud volcanic plume: The persistence of ash in the lower stratosphere, Journal of Geophysical Research: Atmospheres, 121, 11,104–11,118, https://doi.org/10.1002/2016JD025344, URL https://agupubs.onlinelibrary.wiley.com/doi/abs/10.1002/2016JD025344, 2016.

Woods, D. C. and Chuan, R. L.: Size-specific composition of aerosols in the El Chichon volcanic cloud, Geophysical Research Letters, 10, 1041–1044, https://doi.org/https://doi.org/10.1029/GL010i011p01041, URL https://agupubs.onlinelibrary.wiley.com/doi/abs/10.1029/GL010i011p01041, 1983.

Zhu, Y., Toon, O. B., Jensen, E. J., Bardeen, C. G., Mills, M. J., Tolbert, M. A., Yu, P., and Woods, S.: Persisting volcanic ash particles impact stratospheric $SO_2$ lifetime and aerosol optical properties., Nat Commun, 11, 4526, https://doi.org/10.1038/s41467-020-18352-5, 2020.